# SCALING LAWS FOR ASSOCIATIVE MEMORIES

**Vivien Cabannes**
FAIR, Meta

**Elvis Dohmatob**
FAIR, Meta

**Alberto Bietti**
Flatiron Institute

## ABSTRACT

Learning arguably involves the discovery and memorization of abstract rules. The aim of this paper is to study associative memory mechanisms. Our model is based on high-dimensional matrices consisting of outer products of embeddings, which relates to the inner layers of transformer language models. We derive precise scaling laws with respect to sample size and parameter size, and discuss the statistical efficiency of different estimators, including optimization-based algorithms. We provide extensive numerical experiments to validate and interpret theoretical results, including fine-grained visualizations of the stored memory associations.

## 1 INTRODUCTION

As the scale of large language models (LLMs) keeps increasing, scaling laws have become a crucial tool to empirically assess and predict the behavior of these models when varying the number of parameters and training data (Kaplan et al., 2020; Hoffmann et al., 2022). Despite their practical impact, the underlying phenomena leading to such scaling laws remain poorly understood. A better understanding of such phenomena could guide researchers towards improved models, algorithms, and datasets which may lead to improved scaling laws.

Our study focuses on a simple model that aims to be representative of LLMs in two ways. First, we focus on heavy-tailed data distributions over discrete tokens, a natural assumption for text data (Piantadosi, 2014). Second, we consider associative memory models that store input-output pairs through outer-products of finite-dimensional embeddings, and can be seen as a proxy of the intermediate layers of transformers. Indeed, some transformer layers have been found to behave as key-value memories (Geva et al., 2021; Meng et al., 2022), and more generally outer-product associative memory matrices arise naturally from training dynamics on intermediate weights (Bietti et al., 2023). Beyond simple associative recall, the combination of multiple such associative rules at different layers may lead to certain circuits with rich "reasoning" behaviors based on context (Elhage et al., 2021; Bietti et al., 2023; Michaud et al., 2023). For example, an intermediate layer input token may encode for the topic "linux", leading to an output token that will trigger a specific behavior in the transformer's following layers when processing the token "terminal".

Our contributions are as follows:

- We provide precise statistical rates for outer-product memories with random embeddings, and compare different memory storage schemes in the context of Zipf-distributed data.
- We compare theoretical schemes to the weights learned by various optimization algorithms used in practice, and illustrate the role of different design choices with numerical experiments.

**Related work.** Associative memory models have a long history in the literature on neural computation (Steinbuch, 1961; Willshaw et al., 1969; Longuet-Higgins et al., 1970; Kohonen, 1972; Amari, 1972; Little, 1974; Hopfield, 1982; Smolensky, 1990; Schlag et al., 2021; Valle-Lisboa et al., 2023), though the statistical insights we provide for continuous-values random embeddings and heavy-tailed tokens distributions are new, to the best of our knowledge. Memorization behaviors have drawn a lot of attention recently, and are believed to be an important notion to understand the learning happening in deep neural network (e.g., Sukhbaatar et al., 2019; Feldman, 2020; Feldman & Zhang, 2020; Geva et al., 2021; Wu et al., 2022). Building on memorization and heavy-tailed discrete data, our model bears similarities to the ones of Hutter (2021), Michaud et al. (2023) or Debowski (2023), although we focus on practical models with finite capacity. The discrete nature of

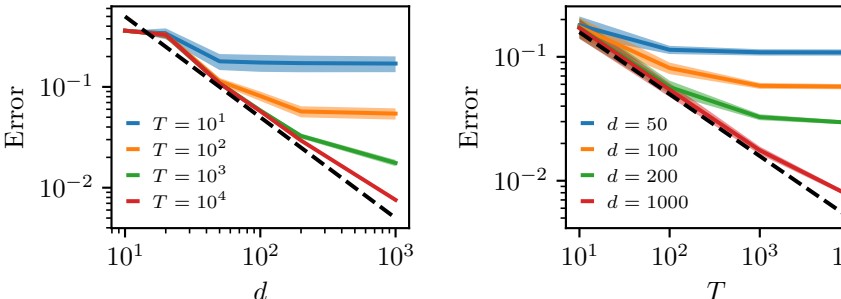

**Figure 1:** Scaling laws with respect to model capacity $d$ (left), respectively the number of data seen $T$ (right), for various numbers of dataset size $T$, respectively various model capacity $d$. This plots validates empirically the theory developed in the paper that proves scaling laws in $\mathcal{E}(f_q) \asymp d^{-\alpha+1} + T^{-1+1/\alpha}$ (dashed lines) under our setting with $\alpha = 2$ (1), (2), (5), and the association scheme (12) with $\rho = 0$ and $P = d/8$. The experiments averaged over 100 runs, standard deviations are shown with solid color.

tokens contrasts with other recent works on scaling laws that have focused on continuous Gaussian inputs (e.g., Bahri et al., 2021; Maloney et al., 2022; Sorscher et al., 2022).

## 2 MODEL FOR ASSOCIATIVE MEMORY

**The data.** In the following, we consider a joint distribution $p \in \Delta_{[N]\times[M]}$ on inputs $x \in [N]$ and outputs $y \in [M]$. The inputs and outputs are respectively assumed to solely take $N$ and $M$ discrete values respectively. For example, $N$ could be the number of potential sequences of fixed word length in the English language, while $M$ would be all the potential words to complete the sequence. Abstractly, $x$ and $y$ will be referred to as tokens. To simplify the study, we assume for now that $y$ is a deterministic function of $x$, i.e., there is no noise in the labels. In consistency with language modeling, we equally assume that $p(x)$ follows a Zipf law. Formally, there exists an parameter $\alpha > 0$, a normalizing constant $C_\alpha$, a permutation $\sigma \in \mathfrak{S}_n$ and a function $f_* : [N] \to [M]$ such that

$$\forall x, y \in [N] \times [M], \qquad p(\sigma(x)) = C_\alpha x^{-\alpha}, \qquad p(y|x) = \mathbf{1}_{y=f_*(x)}. \qquad (1)$$

The distribution $p$ is not known, but has generated $T$ known independent samples $(x_t, y_t)_{t \in [T]} \sim p$. For readability sake, we will assume without restriction that $\sigma$ is the identity (so that $p$ is decreasing).

**The model, and the loss.** The input tokens are embedded into a space $\mathbb{R}^d$ of dimension $d$ through an embedding map $e : [N] \to \mathbb{R}^d$. This space is used for computation purposes. In particular, we focus on the linear transformation parameterized by a matrix $W \in \mathbb{R}^{d\times d}$ mapping $x$ to $We(x)$. This latter vector is mapped back to the output space through an unembedding map $u : [M] \to \mathbb{R}^d$ and the decoding rule

$$f_W(x) = \arg\max_{y\in[M]} u_y^\top W e_x, \qquad W \in \mathbb{R}^{d\times d}, \qquad (2)$$

where $e_x$ and $u_y$ are abbreviations for $e(x)$ and $u(y)$. The model (2) can be seen as analogous to an attention layer where keys $e_x$ are tested against queries $u_y$ through a matrix $W$ before going through a softmax layer, which, when the attention is peaky, identifies to an argmax. It also resembles next-token prediction from an intermediate representation $We_x$, which may itself be the output of an attention block that attends to a token $x$. The matrices $W$ will be expressed as associative memories. Memory of an observed pair $(x, y)$ is represented as an outer product $u_y e_x^\top$. Remembering those with respect to a probability $q \in \Delta_{[N]\times[M]}$ leads to the matrix

$$W_q = \sum_{(x,y)\in[N]\times[M]} q(x,y)u_y e_x^\top, \qquad q \in \Delta_{[N]\times[M]}, \qquad (3)$$

This representation (3) is justified as the predictions (2) are insensitive to modifications of $M$ outside the span of $(u_y e_x^\top)_{x,y}$. In our deterministic setting (1) where one only observes pairs $(x, f_*(x))$, we shall consider the simpler model where[1]

$$W_q = \sum_{x\in[N]} q(x)u_{f_*(x)} e_x^\top, \qquad q \in \Delta_{[N]}. \qquad (4)$$

---

[1]It should be noted that the proof techniques behind Theorem 1 do not break when considering $q = q(x,y)$: both models would lead to similar results, with the case $q = q(x,y)$ being simpler to comprehend.

**Table 1:** Summary of key elements in the study. We are given discrete tokens $x, y$ with deterministic relation $y = f_*(x)$. We embed tokens in $\mathbb{R}^d$, $d$ acts as a "model capacity" parameter. We store association $x \to y$ in the matrix $W$ through a scheme $q$ and recall them through the decoding $f_q$. We will first study the scaling law of the generalization error $\mathcal{E}$ as a function of the number of data $T$, and the model capacity $d$ for different schemes $q$. We will later study the scheme $q$ found by optimization-based algorithms.

| Tokens | Embeddings | Model | Scaling |
|---|---|---|---|
| $y_{i_t} = f_*(x_{i_t})$ | $e_x, u_y \in \mathbb{R}^d$ | $W = \sum_x q(x) u_{f_*(x)} e_x^\top$ | $\mathcal{E}(q) = \mathbb{E}[\mathbf{1}_{f_q(x) \neq f_*(x)}]$ |
| $t \in \{1, 2, \ldots, T\}$ | $e_x \sim \mathcal{N}(0, I)$ | $f_q(x) = \arg\max_y u_y W e_x$ | $\mathcal{E}(q) = F(d, T; q)$ |

**Table 2:** Some insightful provable scaling laws with respect to the model capacity $d$, and the number of data $T$, for two schemes that store associations as (4) and random embeddings.

| Model | Error scaling | Comment |
|---|---|---|
| $q(x) = p(x)$ | $d^{-(\alpha-1)/2\alpha} + T^{-1+1/\alpha}$ | Found with large batches in one step |
| $q(x) = \mathbf{1}_{x \leq d}$ | $d^{-\alpha+1} + T^{-1+1/\alpha}$ | Optimal scaling with random embeddings |

To simplify notations, we will write $f_q$ for $f_{W_q}$ (2). The model $f_q$ is seen as superposing memories since all associations are mixed together in a single matrix. The quality of a mapping $f$ is quantified through the generalization error

$$\mathcal{E}(f) = \mathbb{E}_{(X,Y) \sim p}[\mathbf{1}_{f(X) \neq Y}], \qquad f : [N] \to [M]. \qquad (5)$$

**Which questions are we interested in?** Several questions naturally arise from our model. The first ones are related to scaling laws: how does the error depend on $T$, the number of data? How does it scale with $d$ that encodes for model capacity? The second ones relate to the model itself: how does the error behave for different $q$? What about optimization-based algorithms?

Arguably, the model (2) lays out a simple model to study memorization, which could easily be extended to model more intricate memorization and training behaviors inside a transformer language model. Indeed, memories of the form (4) were found to accurately model the behavior of weight matrices in multi-layer transformers trained by gradient methods on certain tasks (Bietti et al., 2023). Hence, we expect our study to be generalizable to more complex mechanisms in transformers, resulting in rich token interactions to predict the next token in a sequence.

## 3 SCALING LAWS WITH RANDOM EMBEDDINGS

**Why do we make errors?** With a simple deterministic model, one may wonder how can we not learn perfectly the mapping $f_*$. There are two sources of error. One is due to not having enough data to see all the potential association $(x, f_*(x))$, and has already been studied by Hutter (2021). The other one is due to the limited memory capacity of our model, which we illustrate in Figure 2.

**Proposition 1** (Finite data, infinite memory). *Consider a infinite memory model $\hat{f}$, which at time $T$ predicts correctly all $x$ that were seen in the past training, i.e., $x \in \{X_t\}_{t \in [T]}$, where the $(X_t, Y_t)$ were drawn independently at random from a distribution $p \in \Delta_{[N] \times [M]}$. Under the data model the generalization error reads, with respect to the random dataset $\mathcal{D}_T = (X_t, Y_t)_{t \in [T]}$,*

$$\mathbb{E}_{\mathcal{D}_T}[\mathcal{E}(\hat{f})] \asymp T^{-1+1/\alpha}. \qquad (6)$$

*Here, the notation $a \asymp b$ means that there exist two constants $c_1$ and $c_2$ such that $c_1 b \leq a \leq c_2 b$.*

### 3.1 TIGHT ERROR CHARACTERIZATION

The case where one has infinite data but finite memory is intrinsically a deterministic problem. However, characterizing interferences between embeddings and the corresponding generalization error is combinatorial in nature, and is hard to study without specific assumptions on the embeddings $e$ and $u$. A natural choice is to consider them to be random, as is the case at initialization.

**Theorem 1** (Infinite data, finite memory). *Let $M \geq 4$ and $d > 8 \log(M)$. For any memory weight scheme $q : [N] \to \mathbb{R}$, when the embeddings $e_x$ are independent random variables $e_x \sim \mathcal{N}(0, I)$, and the unembeddings are taken uniformly at random on the sphere,*

$$\mathbb{E}_{e,u}[\mathcal{E}(f_q)] \leq \inf_\gamma 2d^{-\gamma} + p\Big(\Big\{x \in [N] \,\Big|\, dq(x)^2 \leq 16c_\gamma\Big(Q_\infty + \frac{8c_\gamma \|q\|_2^2}{d}\Big)\Big\}\Big), \qquad (7)$$

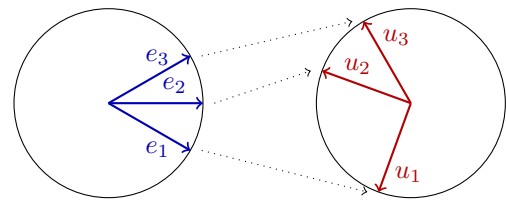

**Figure 2:** Error due to finite memory capacity: the stacking of associative memories in a matrix $W$ may exhibit a pattern $W = \sum_x u_{f_*(x)} e_x^\top$ where three inputs mapped to three different outputs interact in such a way that $u_2^\top W e_1 = e_2^\top e_1 + u_2^\top u_3 e_3^\top e_1 \geq 1 + u_1^\top u_3 e_3^\top e_1 = u_1^\top W e_1$, so that $f_W(x = 1) = 2 \neq 1 = f_*(x = 1)$. In other terms, memory interference may lead to wrong prediction, illustrating the finite capacity of the model $f_W$ (2) to store all data associations.

*where $Q_\infty := \max_y \sum_{x; f_*(x)=y} q(x)^2$, $c_\gamma = \log(M) + \gamma \log(d)$, and $p(\mathcal{X}) = \sum_{x \in \mathcal{X}} p(x)$ denotes the probability of $x$ to belong to $\mathcal{X} \subset [N]$. In terms of lower bound,*

$$\mathbb{E}_{e,u}[\mathcal{E}(f_q)] \geq \frac{1}{20} p(\{x \in [N] \,|\, 3(d+1)q(x)^2 \leq Q_\infty\}). \tag{8}$$

Theorem 1 illustrates how the error made by a scheme $q$ at the input $x$ relates to the ratio between the signal $dq(x)$, provided by the associative memory $u_{f_*(x)} e_x^\top$, and the noise $Q_\infty$, which corresponds to the signal provided by the most competitive class for $y \in [M]$. This is true up to a higher term in $\|q\|^2/d$, which corresponds to a class $y = f_*(x)$ competing against itself when the random embeddings $e_{x'}$ for $x'$ such that $f_*(x') = y$ point in the opposite direction of $e_x$. When $d$ is large and $p$ is regular, $c_\gamma \|q\|_2^2/d$ will be dominated by $Q_\infty$ and the cut-off of $q(x)^2/Q_\infty$ at $32c_\gamma/d$ will behave similarly to a cut-off at $1/d$ up to logarithmic terms. Moreover, when $q$ is chosen independently of $p(y|x)$,[2] one can expect $Q_\infty \approx p_* \|q\|^2$ where $p_* = \max_{y \in [M]} p(y)$. As a consequence, up to constants and logarithmic term, we get

$$\mathbb{E}[\mathcal{E}(f_q)] \approx p(\{x \in [N] \,|\, dq(x)^2 \leq p_* \|q\|^2\}). \tag{9}$$

### 3.2 Memory schemes

Let us now discuss several natural choices for $q$ and compare their corresponding performance. The first naive choice consists in storing all the data seen at time $T$ in memory. It reads

$$\hat{q}_0(x) = \mathbf{1}_{x \in \{X_t\}_{t \in [T]}}, \qquad q_0(x) = 1. \tag{10}$$

Here, $\hat{q}_0$ corresponds to the learned weighted scheme based on the $T$ data, while $q$ denotes an idealized limit when one has infinite data. In the idealized setting $Q_\infty(q_0) = Np_*$ where $p_* := \max_{y \in [M]} p(y)$. From Theorem 1, we deduce that $\mathcal{E}(f_{W_{q_0}})$ will follow two regimes: an overflow regime where $3(d+1) \leq Np_*$ and in essence the memory $W_{q_0}$ is too full to recover any signal in it, and $\mathbb{E}_{e,u}\mathcal{E}(f_{W_{q_0}}) > 1/20$ (8); a infinite memory regime where $d \geq N$ and all associations $e_x u_{f_*(x)}^\top$ can be stored orthogonally to one another, and the error $\mathbb{E}_{e,u}\mathcal{E}(f_{W_{q_0}})$ quantifies the tiny probability that some random inputs embeddings appear to be too correlated.

Equipped with the knowledge that our associative memory model (2) has finite capacity, one may weight memories according to their frequencies, leading to the scheme, for $\rho \geq 0$

$$\hat{q}_\rho(x) = \left( \frac{1}{T} \sum_{t \in [T]} \mathbf{1}_{x = X_t} \right)^\rho, \qquad q_\rho(x) = p(x)^\rho. \tag{11}$$

A better option consists in explicitly limiting the storage of our model with a simple thresholding algorithm

$$\hat{q}_{\rho,[P]}(x) = \hat{p}(x)^\rho \mathbf{1}_{x \in \mathrm{top}_P((x_t)_{t \in [T]})}, \qquad q_{\rho,[P]}(x) = p(x)^\rho \mathbf{1}_{x \in [P]}, \tag{12}$$

where $\mathrm{top}_P((x_t))$ denotes the set made of the $P$ most frequent inputs in the data $(x_t)$.

---

[2]To be more precise, one should actually choose $q(x)$ to be class dependent so to cram in memory as many $x$ as possible for each different class $y = f_*(x)$, ensuring that $y \mapsto \sum_{x; f_*(x)=y} q(x)^2$ is constant with respect to $y$. For simplicity, we will not discuss this behavior that does not change the big picture beyond our exposition.

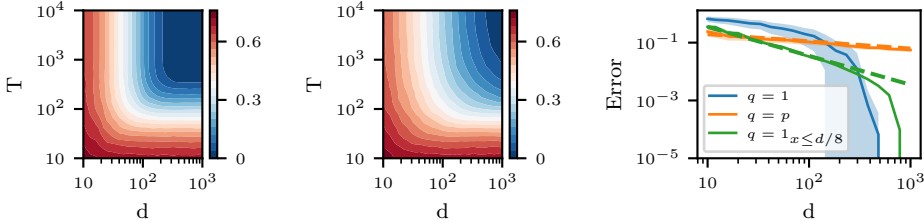

**Figure 3:** Generalization error (5) as a function of $d$ and $T$ for the model (4) averaged over 100 runs. The data follows a Zipf law with $\alpha = 0.5$, $N = 100$, $M = 5$ and $f_*(x) = x \bmod M$. Left: error for $q_0$ (10), either $d$ is too small and there will be memory overflow leading to large error (red area), either it is big enough and with enough data, the error will be null (blue area). Middle: error for $q_1$ (11), for small $d$ and big $T$, it avoid memory overflow allowing a smaller error then $q_0$; however for big $d$ it does not allocated enough memory to rare association, leading to a bigger error. Those results can be interpreted mechanistically by looking at the corresponding memory matrices (see Figure 10). Right: Generalization error when $T = +\infty$, $N = 100$ and $\alpha = 2$: the scheme $q_0$ leads to a zero-one type of plot where if $d < N$ the error is high, and if $d > N$ the error decreases fast to zero (in blue); the scheme $q_1$ leads to an error decreasing in $d^{-(\alpha-1)/2\alpha} = d^{-1/4}$ as predicted by theory (in orange); the scheme $q_{0,P}$ (12) with $P = d/8$, decreases in $d^{-(\alpha-1)} = d^{-1}$ until reaching the tipping point when $d/8 > N$ (in green).

**Proposition 2** (Without thresholding). *Let $p$ be an $\alpha$-Zipf distribution (1). For $\rho > 0$, the performance of $f_\rho := f_{q_\rho}$ (11) is, up to poly-logarithm factors and constants that depends on both $\rho$ and $\alpha$,*

$$\mathbb{E}_{e,u}\mathcal{E}(f_\rho) \overset{(\log)}{\asymp} \left(\frac{d}{\varphi(N)}\right)^{-(\alpha-1)/2\rho\alpha}, \quad where \quad \varphi(N) = \begin{cases} 1 & if\ 2\rho\alpha > 1 \\ \log(N) & if\ 2\rho\alpha = 1 \\ N^{1-2\rho\alpha} & if\ 2\rho\alpha < 1 \end{cases}. \quad (13)$$

*In particular, when $\rho = 1$, $\mathbb{E}_{e,u}\mathcal{E}(f_0)$ scales in $d^{-(\alpha-1)/2\alpha}$. In the limit where $\rho = 0$, $\mathbb{E}_{e,u}\mathcal{E}(f_0)$ can be understood as $(d/N)^{-\infty}$ which will go to zero if and only if $d$ is bigger than $N$.*

**Proposition 3** (With thresholding). *Assume that $p(x)$ follows a $\alpha$-Zipf law (1) with $N = +\infty$. For $\rho \geq 0$, setting $P \simeq d^{1/(2\alpha\rho+1)}$, the error made by the memory scheme (12) scales as*

$$\mathbb{E}_{e,u}\mathcal{E}(f_\rho) \overset{(\log)}{\asymp} d^{-(\alpha-1)/(2\rho\alpha+1)}. \quad (14)$$

In particular, when $\rho = 0$ and $P \simeq d$, one gets a scaling in $d^{-\alpha+1}$, which is actually optimal. The fact that this maximum is reached for $P \simeq d$ is reminiscent of Hopfield networks (Hopfield, 1982) which can only store $d/\log(d)$ patterns with a $d$ by $d$ matrix. Similarly, our model stores at most $d$ associations, which, when in presence of a Zipf law, leads to an error scaling in $d^{-(\alpha-1)}$.

**Theorem 2** (Minimax performance). *Assume that $p(x)$ follows a $\alpha$-Zipf law (1) with $N = +\infty$. For any weighting scheme $q$, and $p_* \in (0,1)$, there exists a conditional distribution $p(y|x)$ with $p_* = \max_y p(y)$ such that the error made for the distribution $p$ is lower bounded by*

$$\mathbb{E}_{e,u}\mathcal{E}(f_q) \geq c_\alpha(d+1)^{-\alpha+1} \quad where \quad c_\alpha = \frac{C_\alpha p_*^{\alpha-1}}{20(\alpha+1) \cdot 3^{\alpha-1}}.$$

*Moreover, this performance is reached (up to logarithms factor) by the thresholding algorithm (12) with $P \simeq d/\log(d)$ and $\rho = 0$.*

Finally, we prove that the scaling laws proved for $d$ when $T = +\infty$ and for $T$ when $d = +\infty$ appears jointly when both $d$ and $T$ are finite.

**Proposition 4** (Finite data and finite memory). *For the previous bound with respect to $d$, Proposition 2 and Proposition 3, considering finite data simply adds a term $T^{-1+1/\alpha}$ (up to constants and logarithmic terms), matching the optimal bound of Proposition 1. In particular, (12) with $\rho = 0$ and $P \simeq d/\log(d)$ reaches the optimal scaling in*

$$\mathbb{E}_{e,u,(x_t,y_t)_{t \in [T]}}\mathcal{E}(f_{\hat{q}}) \asymp T^{-1+1/\alpha} + d^{-\alpha+1}. \quad (15)$$

The optimal scaling (15) recovers the law of Hutter (2021) with respect to $T$, and the one of Michaud et al. (2023) with respect to $d$. This is intuitive, since Hutter (2021) assumes memorizing exactly

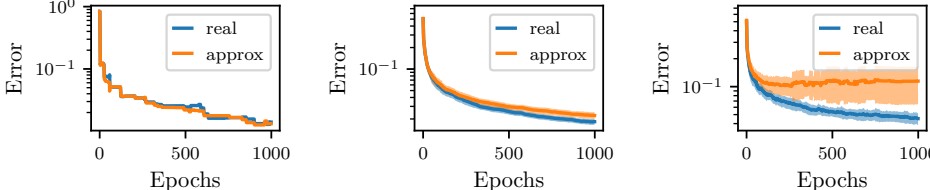

**Figure 4:** Comparison between the error found by optimizing $W$ (2) with SGD on the cross-entropy loss, and its approximation with $q(x)$ (4) and the approximate update rule (20). We consider $N = 100$, $M = 5$, $f_*(x) = x \bmod M$, $\alpha = 2$, and batch size equals one. Left: One run with $d = N = 100$ with $\gamma = 10$. Middle: Average over 100 runs with $d = N = 100$ with $\gamma = 1$. Right: Average when $d = N/10 = 10$ with $\gamma = 1$, which implies that our approximation is not valid anymore. The same results can be obtained for bigger batch sizes as shown in Figure 13.

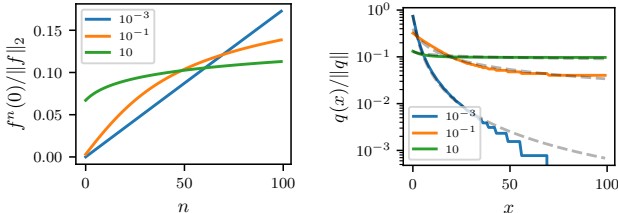

**Figure 5:** Theoretical approximation of the association scheme found with stochastic gradient descent with batch size equals one and fixed learning rates. Left: Plot of $f^n(0)$ as a function of $n$ where $f$ is the effect of one gradient update on $q(x)$ (20). Right: Plot of the resulting $q_\gamma(x)$ when $n_x \propto p(x) \propto (x + 3)^{-\alpha}$ with $\alpha = 2$ and $n_N = 1$. In dashed, we represent $q_\rho$ (11) for $\rho = 0.05$, $\rho = 0.35$ and $\rho = 1$. Those curves map well $q_\gamma$ for $\gamma = 10$, $\gamma = 10^{-1}$ and $\gamma = 10^{-3}$ respectively.

all previously seen data, while each memory could be seen as specifying a "quantum of knowledge" as modeled in Michaud et al. (2023), with $d^{-\alpha+1}$ corresponding to the risk (5) of only storing the most frequent $d$ tokens. However, associative memories can be understood at different level of granularity, and while one may argue that a transformer acts as a big associative memory machine and derives LLMs scaling laws approximations as corollaries, we prefer to understand a transformer as a combination of hidden associative memories as suggested by Sukhbaatar et al. (2019); Geva et al. (2021); Wu et al. (2022); Bietti et al. (2023) among others.

## 4 OPTIMIZATION-BASED MEMORIZATION

This section studies memory schemes privileged by optimization-based algorithms, digging into the training dynamics behind memorization. In terms of relevance, we argue that our model (2) is a proxy for the inner layers of a transformer that memorize patterns before matching them against new data at inference time. As such, we want to understand how different key elements in the training of a transformer influence storage in our memory model.

**Gradient updates.** We consider the cross entropy loss as a surrogate objective to minimize, and study the form of gradient updates on batches of data. Formally, the matrix $W \in \mathbb{R}^{d \times d}$ in (2) is optimized to minimize the loss

$$\mathcal{L}(W) = \mathbb{E}_{(X,Y) \sim p}[\ell(x, y; W)], \qquad \ell(x, y; W) = -u_y^\top W e_x + \log\Big(\sum_{z \in [M]} \exp(u_z^\top W e_x)\Big). \quad (16)$$

The gradient of this loss with respect to $W$ takes the following form, as detailed in Appendix A.10:

$$\nabla_W \ell(x, y; W) = -(1 - p_W(y|x))(u_y - \varepsilon)e_x^\top, \quad \text{with} \quad \varepsilon = \sum_{z \in [M]} p_W(z|x, z \neq y)u_z. \quad (17)$$

where $p_W(y|x) \propto \exp(u_y^\top W e_x)$ are model predictions for the current $W$. For a batch of $n$ data $B = [x_1, \cdots, x_n]$, a gradient update with step size $\gamma_t$ updates $W_t$ as

$$W_{t+1} = W_t - \gamma_t \sum_{x \in B} \nabla_W \ell(x, f_*(x); W_t). \quad (18)$$

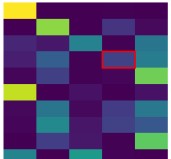 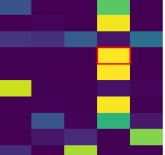 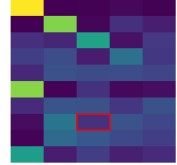 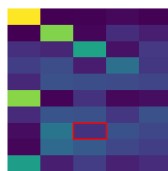

**Figure 6:** Gradient descent dynamics from perspective of the matrix $(u_y^\top W_t e_x)_{y,x} \in \mathbb{R}^{M \times N}$ with $N = 10$, $M = 5$, $\alpha = 1.5$, $f_*(x) = x \bmod 5$, and $d = 5 < N$. A lighter color in the square $(y, x)$ means a higher value of $u_y^\top W e_x$. The optimal $W$ corresponds to two diagonal strips of yellow boxes (see Figure 15). The matrix $W_t$ is updated with stochastic gradient descent with batch size equal to one. From time to time, stochastic gradient descent will hit an association that is not properly stored in memory yet (the red boxes). It will consequently update the weight matrix $W_t \to W_{t+1}$ (side by side pairs) to store it (18). Left pair: update with a big learning rate $\gamma = 10$, whose risk is to erase previous memories (the light colored boxes), similarly to $q_0$ (10). Right pair: update with a small learning rate $\gamma = 10^{-1}$, which will not store rare memory, similarly to $q_\rho$ (11) with large $\rho$.

**Approximation of the updates.** When $p_W(z|x)$ does not change much for all $z \neq f_*(x)$, since $u_z$ were sampled at random in $\mathcal{S}^d$, we expect $\varepsilon$ (17) to concentrate around zero with $\|\varepsilon\|^2 \approx 1/M$, hence to be negligible in front of $u_{f_*(x)}$. As a consequence,

$$\nabla_W \ell(x, f_*(x); W) \approx -(1 - p_W(f_*(x)|x)) u_y e_x^\top. \tag{19}$$

This is notably the case for $W = 0$, random $W$, or if $W$ only stores pairs $(x, f_*(x))$ with $d \gg N$. With the update model above (19), $T$ steps of SGD with batch size one lead to an association scheme of the form (4) with (see Appendix A.11)

$$q_\gamma(x) \approx f^{Tp(x)}(0) = \underbrace{f \circ f \circ \cdots \circ f}_{Tp(x) \text{ times}}(0), \qquad \text{where} \qquad f : x \mapsto x + \frac{\gamma}{1 + M^{-1} \exp(x)}. \tag{20}$$

This equation tells us what form to expect for $q$ for optimization schemes with different hyperparameters. This approximation is shown in Figure 5, and is validated empirically in Figure 4.

**Step size effect.** When $d > N$, the updates approximation (20) and the resulting $q_\gamma$ show how a large learning rate $\gamma$ is beneficial for our problem, in particular when using SGD with batch size one. Interestingly, the same behavior holds in the presence of limited capacity, i.e., $d < N$, although interferences between embeddings (Figure 2) break our approximation (19). In those settings, we resort to numerical simulation to study how optimization manages to rearrange memories. Figure 6 showcases two types of behaviors depending on the size of $\gamma$. *(i)* When the learning rate $\gamma$ is large, associations will be stored easily in memory, but will tend to overwrite previous storage. *(ii)* When the learning rate $\gamma$ is small, associations need to be seen often to build up in the matrix $W$ (4) which will take more time, but will not erase memory. This provides another intuition explanation for why a bigger step size leads to better results on the left of Figure 7. The previous considerations also explain the usefulness of **scheduling** in our simple model, which we illustrate on Figure 11: using a large learning rate enables us to store associations while there is still memory space, while reducing it later in training avoids overwriting previous storage unless an association is highly frequent.

**Batch size effect.** Table 2 recalls how storing associations with $q = 1$ under the model (4) is better than storing them with $q = p$. As such, it suggests that, when processing a finite number of data $T$, smaller batch size is preferable. Intuitively, processing an input $x$ in a batch will reweight it by its frequency $p(x)$, while processing it by itself will update $W$ similarly to setting $q_\gamma(x) = 1$ if $x$ has not been already seen. Indeed, in the large batch limit where $|B| \to +\infty$, one batch update corresponds to a population gradient update, which when $p_W \ll 1$ assimilates to $\nabla_W \mathcal{L}(W) \approx -\sum_x p(x) u_{f_*(x)} e_x^\top$. This contrasts with many small batch updates that rather lead to an association scheme akin to (4) with $q = 1$. In support of this line of reasoning, Figure 7 (middle) illustrates the benefits of splitting the descent with many steps, with a small batch size and large step size.

### 4.1 PRACTICAL CONSIDERATIONS

In order to optimize our simple model the fastest, we have seen the usefulness of large step size and small batch size. However, for large transformers such design choices are impractical. First, large step sizes may lead to instability in realistic models (Gilmer et al., 2021). Second, in order to reduce training time and improve hardware efficiency, one should process large batches (Smith et al., 2018).

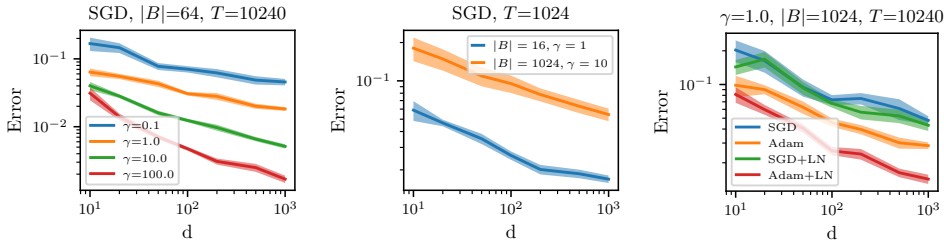

**Figure 7:** Effect of step size, batch size, layer-norm and Adam (with $\beta_1 = \beta_2 = 0$, which corresponds to SignGD). All the experiments are conducted with $N = 100$, $M = 5$, $\alpha = 2$, $f_*(x) = x \bmod M$, averaged over ten runs. We initialized parameters and rescale learning rates to ensure maximal feature updates, as explained in Appendix B.1. To avoid confounders, we scale $\gamma$ on the middle plot for the variance of the gradient updates to be independent of the batch size.

**Adam.** We have seen before how the update of SGD with large batch can be approximated with

$$\gamma_t^{-1}(W_{t+1} - W_{t-1}) = \sum_{x \in B}(1 - p_W(f_*(x)|x))u_{f_*(x)}e_x^\top \approx \sum_{x \in \mathbb{N}}|B|(1 - p_W(f_*(x)|x))p(x)u_{f_*(x)}e_x^\top.$$

Those naive updates would lead to a model that resembles (4) with $q = p^\rho$ for $\rho \approx 1$ (11). In concordance with previous research on the matter (Zhang et al., 2020; Kunstner et al., 2023), we found Adam to be helpful in our setup as well, see Figure 7 (right). In first order approximation, Adam is approximated as signSGD (Balles & Hennig, 2018). Arguably, this introduces a normalization effect to the gradient, helping to reach the saturation phase of $n \mapsto f^n$ (20) shown on Figure 5, homogenizing the resulting matrix $W$ to behave similarly to $q_1 = 1$, therefore optimizing memory capacity. Experiments to underpin this intuition are reported in Figures 15 and 16 in Appendix B.

**Layer normalization.** Minimizing the cross-entropy loss implies setting $p_W(y|x) = 1$, which will lead to $W$ diverging to infinity and unstable loss gradients. In order to ensure numerical stability, it is natural to rescale the vector $We_x \in \mathbb{R}^d$, especially since what matters for the final prediction $f_W$ is only its direction. This is precisely what layer-norm does, introducing the logit score

$$g_y^{\mathrm{LN}}(x) = \langle u_y, \frac{We_x}{\|We_x\|}\rangle, \qquad \text{instead of} \qquad g_y(x) = u_y^\top We_x.$$

This leads to an added projection on the gradients in (17), as detailed in Appendix A.12, denoting $\bar{W} = W/\|We_x\|$,

$$\nabla_W \ell^{\mathrm{LN}}(x, y; W) = \nabla_W \ell(x, y; \bar{W}) = \frac{1}{\|We_x\|}\left(I - (\bar{W}e_x)(\bar{W}e_x)^\top\right)\nabla_{\bar{W}}\ell(x, y; \bar{W}). \quad (21)$$

We recognize a projection that kills the signal that already aligns with $We_x$. We conjecture that this introduces a clipping effect on the corresponding $q(x)$, optimizing for memory storage, and explaining the good performance observed in the right of Figure 7.

## 4.2 THE BENEFITS OF LEARNING THE EMBEDDINGS

Taking a step back, Theorem 1 implies that our model with $d^2$ parameters, the matrix $W \in \mathbb{R}^{d \times d}$ (4), only memorize about $d/\log(d)$ associations $(e_x, u_y) \in (\mathbb{R}^d)^2$ of size $2d$. Intriguingly, Lemma 1 below states that an exponential number of quasi-orthogonal elements can be put in $\mathbb{R}^d$, an event that actually holds with high probability when embeddings are random, showcasing intrinsic limitations of our "linear" model (2).

**Definition 1** (Quasi-orthogonality). *The family $(u_z)_{z \in [P]}$ with $u_z \in \mathbb{R}^d$ is $\eta$-quasi orthogonal if*

$$\forall\{z, z'\} \subset [P], \qquad |\langle u_z, u_{z'}\rangle| \leq \eta, \qquad \text{and} \qquad \|u_z\| = 1. \quad (22)$$

**Lemma 1.** *For any $d \in \mathbb{N}$ and $P \geq 3$, there exists an embedding $u : [P] \to \mathbb{R}^d$ such that the family $(u_z)_{z \in [P]}$ is $\eta = 2\sqrt{d^{-1}\log(P)}$-quasi orthogonal.*

As a consequence of Lemma 1, the following model

$$f_1(x) = \arg\max_y u_y^\top \sum_{x' \in [P]} u_{f_*(x')}\sigma(e_{x'}^\top e_x - \eta), \quad (23)$$

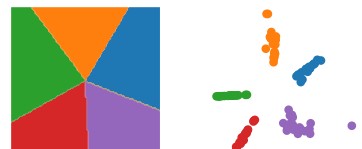

**Figure 8:** Experiments with learned embeddings when $\alpha = 2$, $N = 100$ and $M = 5$ with $y = f_*(x) = x \bmod M$ and $d = 2$. Left: level lines of the function $\mathbb{R}^2 \to [5]; u \mapsto \arg\max_{y \in [5]} u_y^\top u$ with $u_y$ the learned unembedding. Middle: scatter plot of the learned input embeddings $e_x \in \mathbb{R}^2$ for $x \in [N]$ colored accordingly to $f_*(x)$ for $e_x$. It illustrates how the input embeddings match with the output ones, similarly to (24) and Proposition 5. Right: learned input embeddings obtained with $M = 10$, and allowing again a zero generalization error. Reaching a zero error with $d = 2$ greatly contrasts with the condition $d \geq N$ needed to get to a zero generalization error when the embeddings are random.

where $\sigma(x) = x_+$ is the ReLU function, can fit $P = \exp(\eta^2 d/4)$ elements in memory, leading to a scaling in $\mathcal{E}(f_1) \asymp \exp(-(\alpha - 1)\eta^2 d/4)$ when $p(x)$ follows a $\alpha$-Zipf law.[3] Similarly, one could consider higher moments of $e_{x'}^\top e_x$ which has been the basis for modern Hopfield networks (Krotov & Hopfield, 2016; Ramsauer et al., 2021). However, implementing the model (23) requires to keep track of each of the $P$ vectors $e_x \in \mathbb{R}^d$, leading to $Pd$ parameters, in order to only store $P$ associations of size $d$, needing compute that scales with $Pd$ at inference time, rather than just $d^2$,

We also note that when embeddings are learned, it is actually possible to store as many memories as desired, which can be seen from the fact that

$$W = I, \forall y \in [M] \, u_y \in \mathcal{S}^d, e_x = u_{f_*(x)} \qquad \Rightarrow \qquad f_*(x) = \arg\max_y u_y^\top W e_x, \qquad (24)$$

In particular, Figure 8 illustrates the solution found when $d = 2$ by optimization-based algorithms in order to get a zero generalization error on the task of Figure 3 where $M = 5$. Optimizing token embeddings is probably an important element to increase memorization capacity in transformers, although enforcing $e_x = u_{f_*(x)}$ is unrealistic when embeddings are shared over different heads, and the input/output relationships to be learned differ across heads.

## 5 CONCLUSION

This work considers a simple model to study memorization in transformers. Here, memorization is seen as a valuable behavior, the network memorizing useful patterns and association rules. We derive precise scaling laws with respect to both the number of data, and the model size, which plays the role of a model capacity. We quantify the effect of different memorization schemes, illustrating the benefits of uniformly weighted outer products. We leverage these theoretical results to study how different optimization algorithms commonly used for transformers may lead to more efficient memorization. In particular, we showcase the efficacy of small batches and large learning rates, and, under the design constraints resulting from efficient hardware utilization and training stability, the usefulness of Adam and layer normalization.

While our study focuses on simple memorization schemes, it opens up many possible new directions. This includes extending our study to richer models that are closer to transformers, where embeddings, attention and feed-forward layers are trained. This could allow models of scaling laws that capture interactions between tokens, as well as hierarchical behaviors that require multiple layers. We would equally like to leverage our framework for assessing memorization and generalization through clear metrics, and eventually automatically adapt the learning rates as a function of the "free" memory capacity left in a layer.

**Acknowledgements.** The authors would like to thank Léon Bottou as well as Hervé Jégou for many fruitful discussions on memorization mechanisms in transformer language models.

---

[3] This result follows directly from two facts. When input embeddings are chosen at random, the probability that they are not $\eta$-quasi orthogonal is bounded by $P^2 \exp(-d\eta^2/2)$. When input embeddings are $\eta$-quasi orthogonal, $f_1(x) = f_*(x)$ for any $x \in [P]$.

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

# A PROOFS

## A.1 FINITE DATA - PROOF OF PROPOSITION 1

Let us consider the infinite memory model, where an LLM can store in memory all previously seen associations $(x, y)$. At each time $t$, a random positive integer $x$ is drawn from some fixed probability distribution. At time $T$, the LLM would have seen $x_1, \ldots, x_T$ and the associated $f_*(x_t)$, where each $x_t$ is a random positive integer drawn independently from $p$. As such, the LLM would have learned a map $\hat{f}$, that only miscorrects the inputs $x$ which are different from all the $x_t$ for $t \in [T]$. The generalization error reads, with respect to the random dataset $\mathcal{D}_T = (X_t, Y_t)_{t \in [T]}$,

$$\mathbb{E}_{\mathcal{D}_T}[\hat{f}] = \mathbb{P}_{X, \mathcal{D}_T}(X \notin \{X_t\}_{t \in [T]}) = \sum_{x \in [N]} p(x) P_{\mathcal{D}_t}(x \notin \{X_t\}_{t \in [T]}) = \sum_{x \in [N]} p(x)(1 - p(x))^T.$$

Using that $(1 - a)^T = \exp(T \log(1 - a))$ and $2 \log(2)a \le \log(1 + a) \le a$ for any $a \ge -1/2$, we get

$$\sum_{x \in [N]} \mathbf{1}_{p(x) \le 1/2} \cdot p(x) \exp(-2 \log(2)p(x)T) \le \sum_{x=2}^{N} p(x) \exp(-2 \log(2)p(x)T)$$

$$\le \mathbb{E}_{\mathcal{D}_T}[\hat{f}] \le \sum_{x \in [N]} p(x) \exp(-p(x)T).$$

Relating this series to the corresponding integral, we have

$$\int_{x \in [1, N]} p(x) \exp(-2 \log(2)p(x)T)\, \mathrm{d}x - 1/T$$

$$\le \int_{x \in [2, p^{-1}(1/T)]} p(x - 1) \exp(-2 \log(2)p(x - 1)T)\, \mathrm{d}x$$

$$+ \int_{x \in [p^{-1}(1/T), N]} p(x) \exp(-2 \log(2)p(x)T)\, \mathrm{d}x$$

$$\le \sum_{x=2}^{N} p(x) \exp(-2 \log(2)p(x)T) \le \mathbb{E}_{\mathcal{D}_T}[\hat{f}] \le \sum_{x \in [N]} p(x) \exp(-p(x)T)$$

$$\le \int_{x \in [1, N]} p(x) \exp(-2 \log(2)p(x)T)\, \mathrm{d}x + 1/T$$

Letting $N$ goes to infinity, we get the scaling

$$\mathbb{E}_{\mathcal{D}_T}[\hat{f}] \asymp \int_1^\infty p(x) e^{-Tp(x)}\, \mathrm{d}x \pm 1/T. \tag{25}$$

Assuming that $p(x) = Cf(x)$ for some constant $C$, and a smooth strongly decreasing function $f : \mathbb{R}_+ \to \mathbb{R}_+$ such that $\lim_{x \to 0} f(x) = +\infty$, one may consider the change of variable $u = f(x)$, i.e., $x = f^{-1}(u)$. If so,

$$\mathrm{d}x = \mathrm{d}(f^{-1})'(u) = \frac{\mathrm{d}u}{f' \circ f^{-1}(u)}.$$

Hence it holds that

$$\mathbb{E}_{\mathcal{D}_T}[\hat{f}] \asymp \int_1^\infty \frac{-u}{f' \circ f^{-1}(u)} e^{-uT}\, \mathrm{d}u. \tag{26}$$

This relates to the Laplace transform of the function inside the integrand. In particular, one can work out that when $p(x) \propto C_\alpha x^{-\alpha}$, $f^{-1}(u) = u^{-1/\beta}$ from which one can deduce that

$$\int_1^\infty x^{-\alpha} \exp(-Tx^{-\alpha})\, \mathrm{d}x = \frac{\alpha}{\Gamma(\frac{\alpha-1}{\alpha})} T^{-\frac{\alpha-1}{\alpha}},$$

which recovers a result of Hutter (2021).

## A.2 MEMORY CAPACITY - PROOF OF LEMMA 1

The proof of Lemma 1 concerning quasi orthogonal embeddings can be done through a reasoning on random embeddings. Let $(X_i)$ be $P$ independent identically distributed random variables. We are interested in the event where the normalized $(X_i)$ are $\eta$-quasi orthogonal.

$$\mathbb{P}(\cap_{\{i,j\}\subset[P]}\{|\langle X_i, X_j\rangle| \leq \eta\|X_i\|\|X_j\|\}) = 1 - \mathbb{P}(\cup_{\{i,j\}\subset[P]}\{|\langle X_i, X_j\rangle| \geq \eta\|X_i\|\|X_j\|\})$$
$$\geq 1 - \frac{P(P-1)}{2}\mathbb{P}(|\langle X_1, X_2\rangle| \geq \eta\|X_1\|\|X_2\|).$$

If this event can happen, it means that there exists such $\eta$-quasi orthogonal samples. As a consequence, we are looking to maximize $\eta$ such that

$$\mathbb{P}(|\langle X_1, X_2\rangle| \geq \eta\|X_1\|\|X_2\|) < \frac{2}{P(P-1)}. \tag{27}$$

Let us consider $(X_i)$ to be distributed accordingly to a rotation-invariant probability. By symmetry, we have, with $f_1$ denoting the first vector of the canonical basis in $\mathbb{R}^d$,

$$\mathbb{P}(|\langle X_1, X_2\rangle| \geq \eta\|X_1\|\|X_2\|) = \mathbb{P}(|\langle X, f_1\rangle| \geq \eta\|X\|) = \mathbb{P}(|\langle \frac{X}{\|X\|}, f_1\rangle| \geq \eta) \tag{28}$$

By symmetry, the vector $X/\|X\|$ is uniform on the sphere. Using that $\mathbb{P}(|\langle X, f_1\rangle| > \eta) = 2\mathbb{P}(\langle X, f_1\rangle > \eta)$ and

$$\mathbb{P}(|\langle X, f_1\rangle| \geq \eta) = \frac{2}{\text{Vol}(\mathcal{S}^{d-1})}\int_{x\in\mathcal{S}^{d-1}}\mathbf{1}_{x_1\geq\eta}\,dx$$
$$= \frac{2}{\text{Vol}(\mathcal{S}^{d-1})}\int_{x_1=\eta}^{2}\text{Vol}(\sqrt{1-x_1^2}\cdot\mathcal{S}^{d-2})\,dx_1$$
$$= \frac{2\,\text{Vol}(\mathcal{S}^{d-2})}{\text{Vol}(\mathcal{S}^{d-1})}\int_{t=\eta}^{1}(1-t^2)^{\frac{d-1}{2}}\,dt = \frac{2\Gamma(\frac{d}{2}+1)}{\sqrt{\pi}\Gamma(\frac{d}{2}+\frac{1}{2})}\int_{t=\eta}^{1}(1-t^2)^{\frac{d-1}{2}}\,dt.$$

To upper bound this probability, we proceed with

$$\mathbb{P}(|\langle X, f_1\rangle| \geq \eta) = \frac{2\Gamma(\frac{d}{2}+\frac{1}{2})}{\sqrt{\pi}\Gamma(\frac{d}{2}+\frac{1}{2})}\int_{t=\eta}^{1}(1-t^2)^{\frac{d-1}{2}}\,dt \leq \frac{2(\frac{d}{2}+1)^{1/2}}{\sqrt{\pi}}\int_{t=\eta}^{1}\frac{t}{\eta}(1-t^2)^{\frac{d-1}{2}}\,dt$$
$$= \frac{2(\frac{d}{2}+1)^{1/2}}{\sqrt{\pi}}\frac{1}{\eta(d+1)}(1-\eta^2)^{\frac{d+1}{2}} \leq \frac{\sqrt{2}}{\sqrt{\pi}\sqrt{\eta^2 d}}\exp(-\frac{\eta^2 d}{2}).$$

The last inequality follows from the fact that

$$\frac{(d+2)}{(d+1)^2} = \frac{d+1+1}{d+1}\frac{1}{d+1} = \frac{1+\frac{1}{d+1}}{1+\frac{1}{d}}\frac{1}{d} \leq d^{-1},$$

and that for any $x \in (-1, 1)$, the concavity of the logarithm mean that $\log(1+x) \leq x$ hence that

$$(1+x)^n = \exp(n\log(1+x)) \leq \exp(nx).$$

This leads to the following series of implications

$$\exists\,(X_i)\,\eta\text{-quasi orthogonal} \quad \Leftarrow \quad \frac{1}{\sqrt{\pi}}(\frac{\eta^2 d}{2})^{-1/2}\exp(-\frac{\eta^2 d}{2}) \geq \frac{2}{P^2}$$

$$\Leftrightarrow \quad (\frac{\eta^2 d}{2})^{1/2}\exp(\frac{\eta^2 d}{2}) \geq \frac{P^2}{2\sqrt{\pi}}$$

$$\Leftarrow \quad \frac{\eta^2 d}{2} \geq 1 \quad \text{and} \quad \exp(\frac{\eta^2 d}{2}) > \frac{P^2}{2\sqrt{\pi}}$$

$$\Leftarrow \quad \frac{\eta^2 d}{2} \geq 2\log(P) - \log(2\sqrt{\pi}) \geq 1$$

$$\Leftarrow \quad \frac{\eta^2 d}{4} \geq \log(P) \geq \frac{1+\log(2\sqrt{\pi})}{2}.$$

Finally, we have proven the existence of a $\eta$-quasi orthogonal family for

$$\eta \geq \sqrt{4\log(P)d^{-1}}, \qquad \text{as long as} \qquad P \geq 3. \tag{29}$$

## A.3 GENERIC ERROR DECOMPOSITION

The error made by $f_W$ relates to the ordering between the signals $u_{f_*(x)} W e_x^\top$ and the noises $\max_{y \neq f_*(x)} u_y W^\top e_x$.

Let $f_q$ be defined as in the main text. We have the following sequence of equivalence, assuming uniqueness of the argument of the maximum for simplicity,

$$
\begin{aligned}
f_q(x_0) \neq f_*(x_0) \qquad &\Leftrightarrow \qquad \arg\max_{y \in [M]} \sum_{x \in [N]} q(x) e_x^\top e_{x_0} u_{f_*(x)}^\top u_y \neq f_*(x_0) \\
&\Leftrightarrow \qquad \max_{y \in [M]} \sum_{x \in [N]} q(x) e_x^\top e_{x_0} u_{f_*(x)}^\top u_y > \sum_{x \in [N]} q(x) e_x^\top e_{x_0} u_{f_*(x)}^\top u_{f_*(x_0)} \\
&\Leftrightarrow \qquad \max_{y \in [M]} \sum_{x \in [N]} q(x) e_x^\top e_{x_0} u_{f_*(x)}^\top (u_y - u_{f_*(x_0)}) > 0.
\end{aligned}
$$

As a consequence,

$$
\begin{aligned}
\mathcal{E}(f_q) &= \sum_{x_0 \in [N]} p(x_0) \mathbf{1}_{f_q(x_0) \neq f_*(x_0)} \\
&= \sum_{x_o \in [N]} p(x_0) \mathbf{1}_{\max_y \sum_{x \in [N]} q(x) e_x^\top e_{x_0} u_{f_*(x)}^\top (u_y - u_{f_*(x_0)}) > 0}. \tag{30}
\end{aligned}
$$

In other terms, we have proven the following characterization, which holds for any $q$, even if derived from a finite number of data,

$$
\mathcal{E}(f_q) = p(\{x \in [N] \mid \max_y \sum_{x' \in [N]} q(x') e_{x'}^\top e_x \langle u_{f_*(x')}, u_y - u_{f_*(x)} \rangle > 0\}). \tag{30}
$$

## A.4 RANDOM EMBEDDINGS - PROOF OF THEOREM 1

Let us introduce randomness in the model. If each $e_x \sim \mathcal{N}(0, I)$ is actually an independent random Gaussian vector in $\mathbb{R}^d$, we continue our derivation with

$$
\begin{aligned}
\mathbb{E}_e[\mathcal{E}(f_q)] &= \sum_{x_o \in [N]} p(x_0) \mathbb{E}_{e_{x_0}} [\mathbb{P}_{(e_x)_{x \neq x_0}} (f_q(x_0) \neq f_*(x_0) \mid e_{x_0})] \\
&= \sum_{x_o \in [N]} p(x_0) \mathbb{E}_{e_{x_0}} [\mathbb{P}_{(e_x)_{x \neq x_0}} (\max_y \sum_{x \in [N]} q(x) e_x^\top e_{x_0} u_{f_*(x)}^\top (u_y - u_{f_*(x_0)}) > 0 \mid e_{x_0})] \\
&= \sum_{x_o \in [N]} p(x_0) \mathbb{E}_{e_{x_0}} [\mathbb{P}_{(e_x)_{x \neq x_0}} (\max_y Z_y > 0 \mid e_{x_0})].
\end{aligned}
$$

Here, we have introduced the random variables $Z_y$ for $y \neq f_*(x_0)$, inheriting their randomness from $(e \mid e_{x_0})$, and defined by

$$
Z_y = \sum_{x \in [N]} q(x) e_x^\top e_{x_0} u_{f_*(x)}^\top (u_y - u_{f_*(x_0)}). \tag{31}
$$

Those are projections of Gaussian variables, hence are Gaussian. Using the fact that $\mathbb{E}[e_x] = 0$, their mean is

$$
\mu_y := \mathbb{E}[Z_y] = q(x_0) \|e_{x_0}\|^2 u_{f_*(x_0)}^\top (u_y - u_{f_*(x_0)}). \tag{32}
$$

Those variables are correlated. Using the characterization of the mean, we deduce that their covariance reads

$$
\begin{aligned}
\Sigma_{y_1, y_2} &:= \mathbb{E}[(Z_{y_1} - \mathbb{E}[Z_{y_1}])(Z_{y_2} - \mathbb{E}[Z_{y_2}])] \\
&= \sum_{x, x' \neq x_0} q(x) q(x') \mathbb{E}[e_x^\top e_{x_0} e_{x'}^\top e_{x_0}] u_{f_*(x)}^\top (u_{y_1} - u_{f_*(x_0)}) u_{f_*(x')}^\top (u_{y_2} - u_{f_*(x_0)}) \\
&= (u_{y_1} - u_{f_*(x_0)}) (\sum_{x \neq x_0} q(x)^2 e_{x_0}^\top \mathbb{E}[e_x e_x^\top] e_{x_0} u_{f_*(x)} u_{f_*(x)}^\top)(u_{y_2} - u_{f_*(x_0)}) \\
&= (u_{y_1} - u_{f_*(x_0)}) (\sum_{x \neq x_0} q(x)^2 \|e_{x_0}\|^2 u_{f_*(x)} u_{f_*(x)}^\top)(u_{y_2} - u_{f_*(x_0)}).
\end{aligned}
$$

Finally, we obtain the following covariance

$$\Sigma_{y,y'} = \|e_{x_0}\|^2 (u_y - u_{f_*(x_0)})^\top (\sum_{x \neq x_0} q(x)^2 u_{f_*(x)} u_{f_*(x)}^\top)(u_{y'} - u_{f_*(x_0)}). \tag{33}$$

We are left with the computation of the probability that the maximum of the $n$ correlated, non-centered, exchangeable, Gaussian variables $(Z_y)$ is bigger than zero.

**Generic upper bound.** Since we do not care about the scaling with respect to $M$, we proceed with

$$\max_{y \in [M]} \mathbb{P}(Z_y \leq 0) \leq \mathbb{P}(\max Z_y \leq 0) \leq \sum_{y \in [M]} \mathbb{P}(Z_y \leq 0) \leq M \max_{y \in [M]} \mathbb{P}(Z_y \leq 0), \tag{34}$$

which leads to

$$\mathbb{P}_{(e_x)_{x \neq x_0}} (\max_y \sum_{x \in [N]} q(x) e_x^\top e_{x_0} u_{f_*(x)}^\top (u_y - u_{f_*(x_0)}) > 0 | e(x_0))$$

$$\leq \sum_{y \neq f_*(x_0)} \exp(-\mathbf{1}_{\mu_y < 0} \frac{\mu_y^2}{2\Sigma_{y,y}})$$

$$= \sum_{y \neq f_*(x_0)} \exp(-\mathbf{1}_{\langle u_{f_*(x_0)}, u_y - u_{f_*(x_0)}\rangle < 0} \frac{\|e_{x_0}\|^2}{2} \cdot \frac{q(x_0)^2 \langle u_{f_*(x_0)}, u_y - u_{f_*(x_0)}\rangle^2}{\sum_{x \neq x_0} q(x)^2 \langle u_{f_*(x)}, u_y - u_{f_*(x_0)}\rangle^2}).$$

Finally, recognizing a $\chi^2$-variable with $d$ degrees of freedom, for any $a > 0$,

$$\mathbb{E}[\exp(-a\|e_{x_0}\|^2)] = (1 + 2a)^{-d/2} = \exp(-\frac{d}{2}\log(1 + 2a)).$$

This leads to the final bound, with $\chi_{u,x} = \min_{y \in [M]} \mathbf{1}_{\langle u_{f_*(x)}, u_y - u_{f_*(x)}\rangle \leq 0}$.

$$\mathbb{E}_e[\mathcal{E}(f_q)] \leq \sum_{x \in [N]} p(x) \min\{1, \sum_{y \neq f_*(x)} \left(1 + \frac{q(x)^2 \langle u_{f_*(x)}, u_y - u_{f_*(x)}\rangle^2}{\sum_{x' \neq x} q(x')^2 \langle u_{f_*(x')}, u_y - u_{f_*(x)}\rangle^2}\right)^{-\frac{d}{2} \cdot \chi_{u,x}}\}. \tag{35}$$

This holds for any unembedding $u$ and associative weight scheme $q$. In the following, we will assume that the unembedding $u$ are such that $\chi_{u,x} = 1$, which is notably the case when the $u_y$ are normalized (i.e., $u_y \in \mathcal{S}^{d-1}$).

**Matching lower bound.** Going back to (34), one can get a matching lower bound.

$$\mathbb{E}_e[\mathcal{E}(f_q)] \geq \sum_{x \in [N]} p(x) \mathbb{E}_{e_x}[\max_{y \neq f_*(x)} \mathbb{P}(Z_y \leq 0 | e_x)]$$

$$\geq \sum_{x \in [N]} p(x) \max_{y \neq f_*(x)} \mathbb{E}_{e_x}[\mathbb{P}(Z_y \leq 0 | e_x)]$$

$$= \frac{1}{2} \sum_{x \in [N]} p(x)(1 - \max_{y \neq f_*(x)} \mathbb{E}_{e_x}[\text{erf}(\frac{\mu_y}{\sqrt{2\Sigma_{y,y}}})]).$$

To conclude, we need an inequality of anti-concentration for Gaussian variables. In essence, we should distinguish two type of inputs $x \in [N]$:

- the ones where $\mu_y / \Sigma_{y,y}$ will be large enough to store the association $u_{f_*(x)} e_x^\top$, which will lead to an error decreasing exponentially fast;
- the ones where the same ratio is too small and that we should count in the lower bound.

Following this split, one can go for the simple "survival" lower bound

$$\mathbb{E}_e[\mathcal{E}(f_q)] \geq \sup_{t>0} \frac{1 - \mathrm{erf(t)}}{2} \sum_{x_0 \in [N]} p(x_0) \max_{y \neq f_*(x_0)} \mathbb{E}_{e_{x_0}}[\mathbf{1}_{\mu_y^2 \leq 2\Sigma_{y,y}t^2}]$$

$$= \sup_{t>0} \frac{1 - \mathrm{erf(t)}}{2} \sum_{x_0 \in [N]} p(x_0) \max_{y \neq f_*(x_0)} \cdots$$

$$\mathbb{P}_{e_{x_0}}(\|e_{x_0}\|^2 q(x_0)^2 \langle u_{f_*(x_0)}, u_y - u_{f_*(x_0)} \rangle^2 \leq 2t^2 \sum_{x \neq x_0} q(x)^2 \langle u_{f_*(x)}, u_y - u_{f_*(x_0)} \rangle^2).$$

$$\geq \sup_{t,s>0} \frac{1 - \mathrm{erf(t)}}{2} \sum_{x_0 \in [N]} p(x_0) \mathbb{P}_{e_{x_0}}(\|e_{x_0}\|^2 \leq s) \max_{y \neq f_*(x_0)} \cdots$$

$$\mathbf{1}_{sq(x_0)^2 \langle u_{f_*(x_0)}, u_y - u_{f_*(x_0)} \rangle^2 \leq 2t^2 \sum_{x \neq x_0} q(x)^2 \langle u_{f_*(x)}, u_y - u_{f_*(x_0)} \rangle^2}.$$

Without optimizing for constants, taking $t = 1/\sqrt{2}$ and $s = d$, we get the simple "survival bound" that there exists a constant $c$ such that

$$\mathbb{E}_e[\mathcal{E}(f_q)] \geq c \sum_{x \in [N]} p(x) \mathbf{1}_{dq(x)^2 \langle u_{f_*(x)}, u_y - u_{f_*(x)} \rangle^2 \leq \sum_{x' \neq x} q(x')^2 \langle u_{f_*(x')}, u_y - u_{f_*(x)} \rangle^2}. \tag{36}$$

The constant can be computed explicitly as

$$c = \frac{1 - \mathrm{erf}(1/\sqrt{2})}{2} \cdot \mathbb{P}(\|e_{x_0}\|^2 \leq d) > 0.158 \cdot 1/2 = 0.079,$$

where we have used that $\|e_{x_0}\|^2$ is a $\chi^2$-variable with mean $d$ hence smaller median, which implies that $\mathbb{P}(\|e_{x_0}\|^2 < d) > 1/2$.

**Quasi-orthogonal output embeddings.** Let us consider $u : [M] \to \mathbb{R}^d$ such that $(u_y)_{y \in [M]}$ is $\eta$-quasi orthogonal.

*Upper bound.* Going back to (35), we can work out a lower bound with

$$\frac{q(x_0)^2 \langle u_{f_*(x_0)}, u_y - u_{f_*(x_0)} \rangle^2}{\sum_{x \neq x_0} q(x)^2 \langle u_{f_*(x)}, u_y - u_{f_*(x_0)} \rangle^2}$$

$$\geq \frac{q(x_0)^2(1-\eta)^2}{\sum_{x \neq x_0} q(x)^2(\mathbf{1}_{f_*(x)=y}(1+\eta)^2 + \mathbf{1}_{f_*(x)=f_*(x_0)}(1-\eta)^2 + \mathbf{1}_{f_*(x) \notin \{y, f_*(x_0)\}} 4\eta^2)}$$

$$\geq \frac{q(x_0)^2(1-\eta)^2}{4 \sum_{x \neq x_0} q(x)^2(\mathbf{1}_{f_*(x)=y} + \mathbf{1}_{f_*(x)=f_*(x_0)} + \mathbf{1}_{f_*(x) \notin \{y, f_*(x_0)\}} \eta^2)}$$

$$= \frac{1}{4} \frac{q(x_0)^2(1-\eta)^2}{\sum_x q(x)^2((1-\eta^2)\mathbf{1}_{f_*(x) \in \{y, f_*(x_0)\}} + \eta^2) - q(x_0)^2}$$

$$= \frac{1}{4} \frac{q(x_0)^2(1-\eta)^2}{\eta^2 \|q\|^2 + (1-\eta^2) \sum_{x; f_*(x) \in \{y, f_*(x_0)\}} q(x)^2 - q(x_0)^2}$$

$$= \frac{1}{4} \frac{q(x_0)^2(1-\eta)^2}{\eta^2 \|q\|^2 + (1-\eta^2)(Q_y + Q_{f_*(x)}) - q(x_0)^2}.$$

Here, we have used that for the numerator

$$\langle u_{f_*(x_0)}, u_y - u_{f_*(x_0)} \rangle^2 = (\langle u_{f_*(x_0)}, u_y \rangle - 1)^2 \geq (1-\eta)^2,$$

and the same for the term in the denominator (since their ratio cancels out), as well as

$$\langle u_y, u_y - u_{f_*(x_0)} \rangle^2 \leq (1+\eta)^2, \qquad \langle u_{f_*(x)}, u_y - u_{f_*(x_0)} \rangle^2 \leq (2\eta)^2.$$

Moreover, we have introduced

$$Q_y = \sum_{x'; f(x')=y} q(x')^2. \tag{37}$$

Using the fact that $(1+x)^d = \exp(d \log(1+x)) \leq \exp(dx)$, an upper bound directly follows from those derivations,

$$\mathbb{E}_e[\mathcal{E}(f_q)] \leq \sum_{x_0 \in [N]} p(x_0) \min\{1, M \exp\left(-\frac{d(1-\eta)^2}{2} \frac{q(x_0)^2}{4\eta^2 \|q\|_2^2 + 2Q_\infty}\right)\}, \tag{38}$$

where

$$Q_\infty = \max_{y \in [M]} Q_y = \max_{y \in [M]} \sum_{x; f_*(x)=y} q(x)^2. \tag{39}$$

*Matching lower bound.* Similarly, one can work out a lower bound with

$$\frac{q(x_0)^2 \langle u_{f_*(x_0)}, u_y - u_{f_*(x_0)} \rangle^2}{\sum_{x \neq x_0} q(x)^2 \langle u_{f_*(x)}, u_y - u_{f_*(x_0)} \rangle^2} \leq \frac{q(x_0)^2 (1+\eta)^2}{\sum_{x \neq x_0} q(x)^2 (\mathbf{1}_{f_*(x)=y}(1-\eta)^2 + \mathbf{1}_{f_*(x)=f_*(x_0)}(1+\eta)^2)}$$

$$\leq \frac{q(x_0)^2}{\frac{1-\eta}{1+\eta} Q_y + Q_{f_*(x)} - q(x_0)^2}.$$

Combining this with (36), we get the lower bound, with $c = .079$,

$$\mathbb{E}_e[\mathcal{E}(f_q)] \geq c \sum_{x \in [N]} p(x) \mathbf{1}_{(d+1)q(x)^2 \leq \frac{1-\eta}{1+\eta} Q_\infty}. \tag{40}$$

Remark that in the previous lower bound, we have dropped the previous factor $\eta^2 \|q\|^2$ that appears in the upper bound. We expect this term to actually be present in a tighter error characterization. In essence, we expect the embeddings to fill the full space $\mathcal{S}^{d-1}$ so that most of the difference $\langle u_{f_*(x)}, u_y - u_{f_*(x_0)} \rangle^2$ typically behave as $\eta^2$. However, quantifying this precisely is beyond the scope of this paper.

**Random output embeddings.** In the case where the output embeddings are random, we can distinguish two cases. The cases where the embeddings are $\eta$-quasi orthogonal, where one can retake the previous derivations, and the case where they are not, which will have a small probability if $\eta$ is large enough.

Consider $u$ to be random embeddings taking uniformly on the unit sphere. Let us introduce the event

$$E_\eta = \{u \text{ is } \eta\text{-quasi orthogonal}\}.$$

We have seen in the proof of Lemma 1 that

$$1 - \mathbb{P}(E_\eta) \leq \frac{M^2}{2\sqrt{\pi}} \sqrt{\frac{2}{\eta^2 d}} \exp(-\frac{\eta^2 d}{2}). \tag{41}$$

For any random variable $Z$ that is bounded by one, we have the bounds

$$\mathbb{P}(E)\mathbb{E}[Z|E] \leq \mathbb{E}[Z] = (1 - \mathbb{P}(E))\mathbb{E}[Z|\neg E] + \mathbb{P}(E)\mathbb{E}[Z|E] \leq (1 - \mathbb{P}(E)) + \mathbb{E}[Z|E]. \tag{42}$$

The upper bound of Theorem 1 directly follows from plugging (38) and (41) into this last equation

$$\mathbb{E}_{e,u}[\mathcal{E}(f_q)] \leq \frac{M^2}{2\sqrt{\pi}} \sqrt{\frac{2}{\eta^2 d}} \exp(-\frac{\eta^2 d}{2}) + \sum_{x \in [N]} p(x_0) \sum_{y \neq f_*(x_0)} \left(1 + \frac{(1-\eta)^2}{4} \frac{q(x_0)^2}{\|q\|_2^2}\right)^{-\frac{d}{2}}. \tag{43}$$

Since this is true for any $\eta$, one can consider the infimum in the upper bound.

In term of lower bound, retaking (40),

$$\mathbb{E}_{e,u}[\mathcal{E}(f_q)] \geq \sup_{\eta \geq 0} c(1 - \frac{M^2}{2\sqrt{\pi}} \sqrt{\frac{2}{\eta^2 d}} \exp(-\frac{\eta^2 d}{2})) \sum_{x \in [N]} p(x) \mathbf{1}_{(d+1)q(x)^2 \leq 2\frac{1-\eta}{1+\eta} Q_\infty}. \tag{44}$$

In particular, when $d > 8 \log(M)$ one can consider $\eta < 1/2$ such that $\eta^2 d > 4 \log(M)$, which leads to $(\eta - 1)/(\eta + 1) > 1/3$, and, if $M \geq 4$

$$1 - \frac{M^2}{2\sqrt{\pi}} \sqrt{\frac{2}{\eta^2 d}} \exp(-\frac{\eta^2 d}{2}) \geq 1 - \frac{1}{2\sqrt{\pi}} \frac{1}{\sqrt{2 \log(M)}} > 2/3.$$

All together we have proven that, as long as $M \geq 4$ and $d \geq 8 \log(M)$ with $c_1 > .079 \cdot 2/3 > .052$ and $c_2 > 1/3$,

$$\mathbb{E}_{e,u}[\mathcal{E}(f_q)] \geq c_1 \sum_{x \in [N]} p(x) \mathbf{1}_{(d+1)q(x)^2 \leq c_2 Q_\infty}. \tag{45}$$

**Writing upper bounds as survival bounds.** Until now, we have written the upper bounds as the sum of exponential (38) and the lower bounds as a sum of missed associations (45), which we called "survival" bound. In order to best read how tight our characterization is, one can rewrite the upper bounds as survival bounds. In particular, as we did in the lower bound, we will dissociate $x$ corresponding to a small exponential and the other ones. Using the fact that the $p(x)$ sum to one, we get, when the output embeddings are $\eta$-quasi orthogonal,

$$\mathbb{E}_e[\mathcal{E}(f_q)] \leq \sum_{x_0 \in [N]} p(x_0) \min\{1, M \exp\big(-\frac{d(1-\eta)^2}{2} \frac{q(x_0)^2}{4\eta^2\|q\|_2^2 + 2Q_\infty}\big)\}$$

$$\leq \sum_{x_0 \in [N]} p(x_0) \inf_{t>0} M \exp\big(-\frac{t(1-\eta)^2}{4}\big) + \mathbf{1}_{dq(x_0)^2 \leq t(2\eta^2\|q\|_2^2 + Q_\infty)}$$

$$\leq \inf_{t>0} \exp\big(-\frac{t(1-\eta)^2}{4} + \log(M)\big) + \sum_{x \in [N]} p(x)\mathbf{1}_{dq(x)^2 \leq t(2\eta^2\|q\|_2^2 + Q_\infty)}.$$

To simplify the bound, consider the constraints

$$\eta^2 \leq Q_\infty/\|q\|_2^2, \qquad \text{and} \qquad \eta < 1/2, \tag{46}$$

we get, using $t = 16(\log(M) + \gamma \log(d))$ for $\gamma > 0$, we get

$$\mathbb{E}_e[\mathcal{E}(f_q)] \leq \inf_{t>0} \exp\big(-\frac{t(1-\eta)^2}{4} + \log(M)\big) + \sum_{x \in [N]} p(x)\mathbf{1}_{dq(x)^2 \leq t(2\eta^2\|q\|_2^2 + Q_\infty)}$$

$$\leq \inf_{t>0} \exp\big(\frac{-t + 16\log(M)}{16}\big) + \sum_{x \in [N]} p(x)\mathbf{1}_{dq(x)^2 \leq 3tQ_\infty}$$

$$\leq \exp(-\gamma \log(d)) + \sum_{x \in [N]} p(x)\mathbf{1}_{dq(x)^2 \leq 48(\log(M) + \gamma \log(d))Q_\infty}.$$

Finally, when the output embedding are $\eta$-quasi orthogonal with $\eta$ satisfying (46), we get

$$\mathbb{E}_e[\mathcal{E}(f_q)] \leq \inf_{\gamma>0} d^{-\gamma} + \sum_{x \in [N]} p(x)\mathbf{1}_{dq(x)^2 \leq 48(\log(M) + \gamma \log(d))Q_\infty}. \tag{47}$$

When the unembeddings are chosen at random, when $d > 8\log(M)$, one can choose $\eta < 1/2$, and (43) is cast as, chosen $d\eta^2 = 4\log(M) + 2\gamma \log(d)$,

$$\mathbb{E}_{e,u}[\mathcal{E}(f_q)] \leq \inf_{\eta,\gamma} \frac{M^2}{2\sqrt{\pi}} \sqrt{\frac{2}{\eta^2 d}} \exp\big(-\frac{\eta^2 d}{2}\big)$$

$$+ d^{-\gamma} + \sum_{x \in [N]} p(x)\mathbf{1}_{dq(x)^2 \leq 16(\log(M) + \gamma \log(d))(2\eta^2\|q\|_2^2 + Q_\infty)}$$

$$\leq \inf_\gamma \frac{d^{-\gamma}}{2\sqrt{\pi}\sqrt{2\log(M) + \gamma \log(d)}}$$

$$+ d^{-\gamma} + \sum_{x \in [N]} p(x)\mathbf{1}_{dq(x)^2 \leq 16(\log(M) + \gamma \log(d))(\frac{8\log(M) + 4\gamma \log(d)}{d}\|q\|_2^2 + Q_\infty)}$$

$$\leq \inf_\gamma 2d^{-\gamma} + \sum_{x \in [N]} p(x)\mathbf{1}_{dq(x)^2 \leq 16(\log(M) + \gamma \log(d))(\frac{8\log(M) + 4\gamma \log(d)}{d}\|q\|_2^2 + Q_\infty)}.$$

Finally, we have shown that when the embeddings are taken at random

$$\mathbb{E}_{e,u}[\mathcal{E}(f_q)] \leq \inf_\gamma 2d^{-\gamma} + \sum_{x \in [N]} p(x)\mathbf{1}_{dq(x)^2 \geq 16(\log(M) + \gamma \log(d))(\frac{8\log(M) + 4\gamma \log(d)}{d}\|q\|_2^2 + Q_\infty)}. \tag{48}$$

### A.5 PROOF OF PROPOSITION 2

When $p(x) \simeq x^{-\alpha}$, $q(x) = p(x)^\rho \simeq x^{-\rho\alpha}$, hence,

$$p(\{x \in [N] \,|\, dq(x)^2 \leq p_*\|q\|^2\}) \simeq p(\{x \in [N] \,|\, x \leq (d\|q\|^{-2})^{1/2\rho\alpha}\}) \simeq (d\|q\|^{-2})^{-(\alpha-1)/2\rho\alpha}.$$

We are left with the computation of $\varphi(N) := \|q\|^2 \simeq \int_1^N q(x)^2 \, dx \simeq \int_1^N x^{-2\rho\alpha} \, dx$. When $2\rho\alpha > 1$, this integral reads $1 - N^{-2\alpha\rho+1}$ which is bounded by one.

## A.6   PROOF OF PROPOSITION 3

When $p(x) \simeq x^{-\alpha}$, $q(x) = \mathbf{1}_{x \in [P]} p(x)^\rho \simeq \mathbf{1}_{x \in [P]} x^{-\rho\alpha}$, we get

$$p(\{x \in [N] \mid dq(x)^2 \leq p_* \|q\|^2\}) = p(\{x \in [P] \mid dq(x)^2 \leq p_* \|q\|^2\}) + p(\{x > P\})$$
$$\simeq \big(\frac{d}{\varphi(P)}\big)^{-(\alpha-1)/2\rho\alpha} + P^{-\alpha+1}.$$

The optimal threshold $P$ is set by equalizing the two terms, which we compute as

$$\big(\frac{d}{\varphi(P)}\big)^{-(\alpha-1)/2\rho\alpha} = P^{-\alpha+1}$$
$$\Leftrightarrow \quad \frac{-\alpha+1}{2\rho\alpha} \log(d) - \frac{-\alpha+1}{2\rho\alpha} \log(P) = (-\alpha+1)\log(P)$$
$$\Leftrightarrow \quad \log(d) - \log(P) = 2\rho\alpha \log(P)$$
$$\Leftrightarrow \quad P = d^{1/(2\rho\alpha+1)}.$$

This choice of $P$ leads to a scaling in, with $f_{\rho,[P]} = f_{q_{\rho,[P]}}$,

$$\mathbb{E}_{e,u}[\mathcal{E}(f_{\rho,[P]})] \stackrel{(\log)}{\asymp} p(\{x \in [N] \mid dq(x)^2 \leq p_* \|q\|^2\}) \simeq P^{-(\alpha-1)} = d^{-(\alpha-1)/(2\rho\alpha+1)}.$$

## A.7   PROOF OF THEOREM 2

The lower bound directly follows from (8) together with $Q_\infty = p_* \|q\|^2$ and the fact that $q$ is invariant to rescaling, so the best we can do is fit as much memories $P$ as we can until reaching $3(d+1) = p_* P$ leading to a scaling in $\int_P^\infty p(x)\,\mathrm{d}x = C_\alpha P^{-\alpha+1}/(\alpha+1)$.

## A.8   PROOF OF PROPOSITION 4

In order to get scaling with both finite data and finite memory simultaneously, we used a simple strategy:

- With high probability $1 - cT^{-1+1/\alpha}$ for some constant $c$, $\hat{q}$ is similar to $q$.
- When $\hat{q}$ is similar to $q$, the scaling with $d$ derived from Theorem 1 is left unchanged by substituting $q$ by $\hat{q}$.

Rather than using a uniform concentration inequality on the full $\hat{q}$, we will proceed individually on each $\hat{q}(x)$. Denoting by $\mathcal{D}_T$ the random dataset of $T$ data, for any sequence of set $(E_x)_{x \in [N]}$ –typically we will choose $E_x = \{\hat{q}(x) > q(x)/2\}$,

$$\mathbb{E}_{u,e,\mathcal{D}_T}[\mathcal{E}(f_{\hat{q}})] = \sum p(x) \mathbb{P}_{u,e,\mathcal{D}_T}(f(x) \neq f_*(x))$$
$$\leq \sum p(x) \mathbb{P}_{u,e,T}(\hat{q} \notin E_x) + \sum p(x) \mathbb{P}_{u,e,T}(f(x) \neq f_*(x) \mid \hat{q} \in E_x).$$

The second term has been worked out before, using that $Q_\infty \leq \|q\|_2^2$

$$\mathbb{P}_{u,e,T}(f(x) \neq f_*(x) \mid \hat{q} \in E_x) \leq \inf_\gamma 2d^{-\gamma} + \mathbb{P}_T(d\hat{q}(x)^2 \leq 16c_\gamma(\hat{Q}_\infty + \frac{8c_\gamma \|\hat{q}\|_2^2}{d}) \mid \hat{q} \in E_x).$$
$$\leq \inf_\gamma 2d^{-\gamma} + \mathbb{P}_T(d\hat{q}(x)^2 \leq c'_\gamma \|\hat{q}\|_2^2 \mid \hat{q} \in E_x),$$

where $c'_\gamma = 16c_\gamma(1 + \frac{8c_\gamma}{d})$.

**Without thresholding.**   Let us first start with the scheme (11), with $\rho > 0$

$$\hat{q}(x) = \big(\frac{1}{T} \sum_{t \in [T]} \mathbf{1}_{x=X_t}\big)^\rho, \qquad q(x) = p(x)^\rho.$$

Using a simplification of Chernoff bound for Bernoulli variables (see e.g., Hoeffding, 1963), we get the probability bound (the randomness being due to the data),

$$\mathbb{P}_T(\hat{q}(x) < \frac{q(x)}{2^{1/\rho}}) = \mathbb{P}_T(\hat{p}(x) < \frac{p(x)}{2}) \leq \exp(-Tp(x)/8).$$

As a consequence, reusing the proof of Proposition 1, when $p$ follows a Zipf law (1),

$$\begin{aligned}
\mathbb{E}[\mathcal{E}(f_{\hat{q}})] &= \sum p(x)\mathbb{P}(f(x) \neq f_*(x)) \\
&\leq \sum p(x)\exp(-Tp(x)/8) + \sum p(x)\mathbb{P}(f(x) \neq f_*(x) \mid \hat{q}(x) > q(x)/2^{1/\rho}) \\
&\lesssim T^{-1+1/\alpha} + \sum p(x)\mathbb{P}(f(x) \neq f_*(x) \mid \hat{q}(x) > q(x)/2^{1/\rho}).
\end{aligned}$$

We are left with the computation of the second term, denote $c_\rho = 2^{-1/\rho}$, we have

$$\mathbb{E}_{u,e}\mathbb{P}_T(f(x) \neq f_*(x) \mid \hat{q}(x) > c_\rho q(x)) \leq \inf_\gamma 2d^{-\gamma} + \mathbb{P}_T(d\hat{q}(x)^2 \leq c'_\gamma\|\hat{q}\|_2^2 \mid \hat{q} \geq q(x)/2).$$

By definition of $\hat{q}$, together with Jensen's inequality when $\rho \leq 1/2$

$$\frac{1}{N} = \frac{1}{N}\sum_{x\in[N]}(q(x)^2)^{1/2\rho} \geq (\frac{1}{N}\|q\|_2^2)^{1/2\rho},$$

hence $\|q\|^2 \leq N^{1-2\rho}$. When $\rho > 1/2$, the worst value of $\|q\|$ is when all the mass is concentrated on one $q(x')$, in which case $\|q\|^2 \leq 1$. With the corresponding $\psi(N)$, we get

$$\mathbb{E}_{u,e}\mathbb{P}_T(f(x) \neq f_*(x) \mid \hat{q}(x) > c_\rho q(x)) \leq \inf_\gamma 2d^{-\gamma} + \mathbf{1}_{dc_\rho^2 q(x)^2 \leq c'_\gamma \varphi(N)}.$$

Finally, reusing the proof of Proposition 2, and hiding logarithmic factors,

$$\begin{aligned}
\mathbb{E}[\mathcal{E}(f_{\hat{q}})] &= \sum p(x)\mathbb{P}(f(x) \neq f_*(x)) \\
&\lesssim T^{-1+1/\alpha} + \inf_\gamma 2d^{-\gamma} + p(\{x \mid dc_\rho^2 q(x)^2 \leq c'_\gamma \psi(N)\}). \\
&\lesssim T^{-1+1/\alpha} + (\frac{d}{\psi(N)})^{-(\alpha-1)/2\rho\alpha}.
\end{aligned}$$

The case $\rho = 0$, can be easily treated by considering an error if and only if the number of seen elements $|\{x_t \mid t \in [T]\}|$ is smaller than $d$.

**With thresholding.** Let us now consider the thresholding scheme (12), with $P \in \mathbb{N}$ and $\rho \geq 0$

$$\hat{q}(x) = \hat{p}(x)^\rho \mathbf{1}_{x\in\mathrm{top}_P((x_t)_{t\in[T]})}, \qquad q(x) = p(x)^\rho \mathbf{1}_{x\in[P]}.$$

We basically proceed with the same technique but with the event $E_x$ the probability that $x$ belongs to the top $P$ of the empirical frequencies. When dealing with a binomial distribution, one can enumerate all possible outcomes for the empirical frequencies. For a template $a \in \Delta_{[N]}$, we said that a sequence $(x_t)$ is of type $a$ if its empirical frequency is equal to $a$,

$$\mathcal{T}(a) = \{(x_t) \in [N]^T \mid \quad \forall x \in [N], \sum_{t\in[T]}\mathbf{1}_{x_t=x} = Ta(x)\}.$$

Some enumeration arguments that can be found in Cover & Thomas (1991, Chapter 11) leads to

$$\mathbb{P}_{\mathcal{D}_T}((x_t) \in \mathcal{T}(a)) = |\mathcal{T}(a)|\exp(-T(H(a) + D_{\mathrm{KL}}(a\|p))) \leq \exp(-T \cdot D_{\mathrm{KL}}(a\|p))).$$

Hence, the probability that $x$ does not belong to the top $P$ of the empirical frequencies of $(x_t)$ is bounded by

$$\mathbb{P}_{\mathcal{D}_T}(x \notin \mathrm{top}_P(x_t) \in \mathcal{T}(a)) \leq \sum_{a\in\mathcal{A}}\exp(-T \cdot D_{\mathrm{KL}}(a\|p))),$$

where $\mathcal{A}$ is the set of all templates $a$ where $x$ is not in the top $P$ of $(a(x'))_{x'\in[N]}$. With $T$ samples over $N$ elements there are at most $(N+1)^T$ different type templates, hence

$$\sum_{a\in\mathcal{A}}\exp(c_a \cdot T) \leq (T+1)^N \sup_{a\in\mathcal{A}}\exp(c_a \cdot T) = \sup_{a\in\mathcal{A}}\exp(c_a \cdot T + N\log(T+1)).$$

As a consequence,

$$\mathbb{P}_{\mathcal{D}_T}(x \notin \mathrm{top}_P(x_t) \in \mathcal{T}(a)) \leq \sup_{a\in\mathcal{A}}\exp(-T \cdot D_{\mathrm{KL}}(a\|p)) + N\log(T+1))$$

Now, it is actually to possible to remove the $N \log(T + 1)$ in the exponential and extends this type of result to generic Polish spaces (see, e.g. Dinwoodie, 1992).

$$\mathbb{P}_{\mathcal{D}_T}(x \notin \text{top}_P(x_t) \in \mathcal{T}(a)) \leq \sup_{a \in \mathcal{A}} \exp(-T \cdot D_{\text{KL}}(a\|p)))$$

We are left with the computation of the "information projection distance" between $p$ and the set of distribution where $x$ does not belong to the top $P$. In order to get $x$ out of the top $P$ of $p$ one should switch $p(x)$ with $p(P)$, which leads to (without caring for exact constants)

$$D_{\text{KL}}(p'\|p) \simeq p(x) \log(p(x)/p(P)) + p(P) \log(p(P)/p(x)) = (p(x) - p(P)) \log(p(x)/p(P))$$

When considering $x < P/2$ and $p$ following a Zipf law we get

$$D_{\text{KL}}(p'\|p) \gtrsim (p(x) - p(2x)) \log(p(P/2)/p(P)) \geq c_\alpha x^{-\alpha}(1 - 2^{-\alpha})\alpha \log(2) = c'_\alpha p(x)$$

where $c'_\alpha = c_\alpha(1 - 2^{-\alpha})\alpha \log(2)$. As a consequence, for any $P \in \mathbb{N}$,

$$\mathbb{E}_{\mathcal{D}_T}[\mathcal{E}(f_{\hat{q}})] \leq c_0 P^{-\alpha+1} + \sum_{x \in [P/2]} p(x)\mathbb{P}(f(x) \neq f_*(x)).$$

$$\leq c_0 P^{-\alpha+1} + \sum_{x \in [P/2]} p(x)(\exp(-Tc'_\alpha p(x)) + \mathbb{P}(f(x) \neq f_*(x) \,|\, x \in \text{top}_P((x_t))))$$

$$\leq c_0 P^{-\alpha+1} + \exp(-2^\alpha Tc'_\alpha P^{-\alpha}) + \sum_{x \in [P/2]} p(x)\mathbb{P}(f(x) \neq f_*(x) \,|\, x \in \text{top}_P((x_t)).$$

When $\rho = 0$, setting $P = \min(c_1 d, T^{-1/\alpha}/\log(T))$ with $c_1$ chosen so that all $x$ stored in memory lead to $f_*(x) = f(x)$ gives to the right scaling with both $T$ and $d$: up to logarithmic factors,

$$\mathbb{E}[\mathcal{E}(f_{\hat{q}})] \lesssim d^{-\alpha+1} + T^{-1+1/\alpha} + \exp(-c_3 \log(T)^\alpha).$$

Because $\alpha > 1$, the last term decreases faster than any polynomial power of $T$, hence ends up being negligible in front of $T^{-1+1/\alpha}$.

For the case $\rho \in (0, 1]$ one can dissociate two events: the event where $x$ belongs to the top $P/2$ empirical frequencies; the event where $\hat{p}(x) > p(x)/2$; and conclude with similar derivations as precedently

$$\mathbb{E}[\mathcal{E}(f_{\hat{q}})] \leq c_0 P^{-\alpha+1} + \exp(-2^\alpha Tc'_\alpha P^{-\alpha}) + c_4 T^{-1+1/\alpha}$$
$$+ \sum_{x \in [P/2]} p(x)\mathbb{P}(f(x) \neq f_*(x) \,|\, x \in \text{top}_P((x_t)), \hat{p}(x) > p(x)/2).$$

Retaking previous arguments leads to the same scalings as the ones of Proposition 3 with respect to $d$ and a scaling in $T^{-1+1/\alpha}$ with respect to $T$. This ends the proof of the mixed scaling with both finite data and finite memory capacity.

## A.9 LEARNING THE INPUTS EMBEDDINGS

In instances where the embeddings are learned within the linear model (2), one may optimize them by merging all input token embeddings that are associated with the same output, which is what we actually observed in practice in Figure 8. Proposition 5 captures the resulting theoretical performance.

**Proposition 5** (Improvement for learned inputs embeddings)**.** *Let the input embeddings be set to $e_x = u_{f_*(x)}$. Assume without restrictions that $p(y)$ is decreasing with $y$. Consider the unembeddings where $(u_y)_{y \in [P]}$ are $\eta$-quasi orthogonal, and $u_y = 0$ if $y$ is not among the $P$-th most frequent classes. Let $q_0 \in \mathbb{R}^N$, and set $q \in \mathbb{R}^M$ as $q(y) = \sum_{x;f_*(x)=y} q_0(x)$, then*

$$\mathcal{E}(f_{W_{q_0}}) \leq p(\{x \,|\, \mathbf{1}_{f_*(x) \notin [P]}q(f_*(x)) < 2\eta\|q\|_\infty + 2\eta^2\|q\|_1\}). \tag{49}$$

*In particular, it is possible to consider a thresholding associative scheme $q_*$ such that, if $y$ follows a Zipf law $p(y) = C_\beta y^{-\beta}$, $\mathcal{E}(f_{W_{q_*}}) = O((d/\log(d))^{-\beta+1})$.*

Proposition 5 shows that when learning the input embeddings one can expect to replace the scaling in $d^{-\alpha+1}$ that depends on the law of $x$, by a scaling that depends on the law of $y$. It illustrates the usefulness to learn embeddings when the law of $x$ is well factored by the law of $y$. This is typically the case when $x$ are news articles associated with a few topics $y$.

*Proof.* When $e$ can be optimized, it is natural to set $e_x$ to be a constant for all $x$ that are associated with the same output. Let $q_0 \in \Delta_{[N]}$ be an associative scheme,

$$\mathbf{1}_{f_{W_{q_0}}(x_0) \neq f_*(x_0)} = \mathbf{1}_{\max_{y \neq f^*(x_0)} \sum_{x \in [N]} q_0(x) e_{x_0}^\top e_x u_x^\top (u_y - u_{f_*(x_0)}) > 0}.$$

In order to lower the probability, one wants to minimize the left expression, which leads to the will to maximize $e_{x_0}^\top e_x u_{f_*(x)}^\top u_{f_*(x_0)}$. This can be done by setting

$$\forall\, x, x' \in [N], \qquad e_x^\top e_{x'} = u_{f_*(x)}^\top u_{f_*(x')}. \tag{50}$$

Such an isometry can be built by setting $e_x = u_{f_*(x)}$, leading to the new characterization

$$\mathbf{1}_{f_{W_{q_0}}(x_0) \neq f_*(x_0)} = \mathbf{1}_{\max_{y \neq y_0} \sum_{z \in [M]} q_0(z) u_{y_0}^\top u_z u_z^\top (u_y - u_{y_0}) > 0},$$

where $y_0 = f_*(x_0)$ and

$$q(y) := \sum_{x;\, f_*(x) = y} q_0(x). \tag{51}$$

When $u$ are $\eta$-quasi orthogonal for its first $P$ values and set to zero otherwise, we have

$$\sum_{z \in [M]} q(z) u_{y_0}^\top u_z u_z^\top (u_y - u_{y_0}) = q(y_0)(u_{y_0}^\top u_y - 1) + q(y)(u_{y_0}^\top u_y - (u_{y_0}^\top u_y)^2)$$

$$+ \sum_{z \in [M] \neq \{y, y_0\}} q(z) u_{y_0}^\top u_z (u_z^\top u_y - u_z^\top u_{y_0})$$

$$\leq -q(y_0) + |q(y_0)|\eta + |q(y)|\eta + \sum_{z \in [M] \neq \{y, y_0\}} |q(z)|\eta(\eta + \eta)$$

$$\leq -q(y_0) + 2\eta \sup_{z \neq y_0} |q(z)| + 2\eta^2 \sum_{z \in [M]} |q(z)|$$

$$= -q(y_0) + 2\eta \|q\|_\infty + 2\eta^2 \|q\|_1.$$

As a consequence, we get

$$\mathcal{E}(f_{W_{q_0}}) \leq \sum_{x \in [N]} p(x) \mathbf{1}_{q(f_*(x)) \leq 2\eta \|q\|_\infty + 2\eta^2 \|q\|_1}.$$

Using that, for any $A : [M] \to \mathbb{R}^d$, $\sum_x p(x) A(f_*(x)) = \sum_x \sum_y p(x, y) A(y) = \sum_y p(y) A(y)$,

$$\mathcal{E}(f_{W_{q_0}}) \leq \sum_{y \in [M]} p(y) \mathbf{1}_{q(y) \leq 2\eta \|q\|_\infty + 2\eta^2 \|q\|_1}. \tag{52}$$

Note that when the embeddings $u$ are chosen uniformly at random on the sphere, and $d > 4 \log(M)$, a similar bound will hold up to an extra higher-order term as seen in the proof of Theorem 1.

When $u$ is defined to be zero on $[M] \setminus [P]$, and only $\eta$-quasi orthogonal for $(u_y)_{y \in [P]}$, the same characterization holds with

$$\mathcal{E}(f_{W_{q_0}}) \leq \sum_{y \in [M] \setminus [P]} p(y) + \sum_{y \in [P]} p(y) \mathbf{1}_{q(y) \leq 2\eta \|q\|_\infty + 2\eta^2 \|q\|_1}. \tag{53}$$

Finally, if $\eta^2$ is set to $1/4P$, and $q_* = \mathbf{1}_{y \in [P]}$, we get the upper bound

$$\mathcal{E}(f_{W_{q_0}}) \leq p(\{x \,|\, f_*(x) > P\}).$$

The best $P$ that one can consider is that such $d/4P = \eta^2 d = 4 \log(P)$. Setting $P = d/16 \log(d)$, and bounding $\sum_{y > P} y^{-\beta} \leq \int_P^\infty t^{-\beta} \, \mathrm{d}t$ ends the proof. $\qquad\square$

### A.9.1 DISCUSSION ON COMPENSATION MECHANISMS

When optimizing the embeddings, one may turn the negative interference mechanisms illustrated in Figure 2 into positive ones.

Assume that $e_x = u_{f_*(x)}$, our model (4) become, denoting $u_{f_*(x)} = u_0$ for simplicity,

$$f(x) = \arg\max_{y \in [M]} u_y^\top W u_0; \qquad W = \sum_{y' \in [M]} q(y') u_{y'} u_{y'}^\top. \tag{54}$$

Similarly as before an error is made when

$$\max_{y \in [M]} \sum_{y' \in [M]} q(y')(u_y - u_0)^\top u_{y'} u_{y'}^\top u_0 > 0. \tag{55}$$

When the output embeddings are learned, one can optimize them to induce compensation mechanisms. For example, when $M = 3$, and $y_1$ is competing when $y_0 = f_*(x)$ as the argmax of (54) due to a large storage of $q(y_1)$ compared to $q(y_0)$, one could benefit of $q(y_2)$ to ensure that

$$q(y_0)(u_1^\top u_0 - 1) + q(y_1)(1 - u_1^\top u_0)u_1^\top u_0 + q(y_2)(u_1 - u_0)^\top u_2 u_2^\top u_0$$
$$< 0 < q(y_0)(u_1^\top u_0 - 1) + q(y_1)(1 - u_1^\top u_0)u_1^\top u_0.$$

In this situation, the score $u_0^\top W e_x$ of $y_0$ would be higher then $u_{y_1}^\top W e_x$ ensuring that we do not make an error when predicting $f(x)$ (54).

We refer the interested reader to Elhage et al. (2022) for related investigation.

### A.10 LOSS GRADIENT

The cross-entropy loss is written as

$$\ell((x, y, W)) = -\log\left(\frac{\exp(u_y^\top W e_x)}{\sum_{z \in [M]} \exp(u_z^\top W e_x)}\right) = -u_y^\top W e_x + \log\left(\sum_{z \in [M]} \exp(u_z^\top W e_x)\right).$$

Hence stochastic gradient descent will update the matrix $W$ by adding terms of the form

$$\partial_W \ell((x, y), W) = -u_y e_x^\top + \frac{\sum_{z \in [M]} \exp(u_z^\top W e_x) u_z e_x^\top}{\sum_{y \in [M]} \exp(u_y^\top W e_x)}$$

$$= -u_y e_x^\top + \sum_{z \in [M]} p_W(z|x) u_z e_x^\top$$

$$= -(1 - p_W(y|x)) u_y e_x^\top + \sum_{z \neq y} p_W(z|x) u_z e_x^\top$$

$$= -(1 - p_W(y|x))(u_y e_x^\top - \sum_{z \neq y} \frac{p_W(z|x)}{1 - p_W(y|x)} u_z e_x^\top).$$

Note that $p_W(z|x)/(1 - p_W(y|x))$ corresponds the the probability of the $z$ conditioned with respect to $x$ under the event that $z$ is not $y$, formally

$$\frac{p_W(z|x)}{1 - p_W(y|x)} = p(z|x, z \neq y).$$

Finally,

$$\partial_W \ell((x, y), W) = -(1 - p_W(y|x))(u_y e_x^\top - \sum_{z \neq y} p_W(z|x, z \neq y) u_z e_x^\top)$$

$$= -(1 - p_W(y|x))(u_y e_x^\top - \mathbb{E}_{z \sim p_W}[u_z|x, z \neq y] e_x^\top).$$

While, it is clear that the model (4) does not describe the solution found by cross entropy, one might hope that the term $\mathbb{E}[u_z] e_x^\top$ will somewhat cancel themselves out and be an order of magnitude smaller than the leading term $u_y e_x^\top$.

### A.11 APPROXIMATE UPDATES

The formula (20) is justified by the fact that a matrix $W_t = W_{q_t}$ will lead to an update (18) at time $t$ according to the rule (19), assuming $\exp(u_z W e_x) \approx 1$ for any $z \neq f_*(x)$,

$$q_{t+1}(x) - q_t(x) = \mathbf{1}_{x_t = x} \gamma \cdot (1 - p_{W_{q_t}}(f_*(x)|x)) \approx \frac{\mathbf{1}_{x_t = x} \gamma}{1 + (M-1)^{-1} \exp(q_t(x))},$$

together with the fact that $x$ will be seen $Tp(x)$ times on average in $T$ samples.

Similarly, very large batch size $b = |B|$ and $T/b$ update steps, each $x$ will appear in each batch about $bp(x)$ times, which leads to the rough approximation

$$q_{\gamma,b}(x) = f^{T/b}(0) = \underbrace{f \circ f \circ \cdots \circ f}_{T/b \text{ times}}(0), \qquad \text{where} \qquad f : x \mapsto x + \frac{\gamma bp(x)}{1 + M^{-1} \exp(x)}. \quad (56)$$

In practice, we can approximate the effect of a batch by counting how many times $x$ was in this batch and setting $bp(x)$ to be the exact count, which will lead to tighter approximation. This is this approximation that we plot on Figure 13.

### A.12 GRADIENT FOR LAYER NORM

Let $x \in [N]$, $y \in [M]$ and $W \in \mathbb{R}^{d \times d}$. When processing the input $x$, layer norm adds a normalization layer

$$f : W \mapsto \bar{W} = \frac{W}{\|W e_x\|}.$$

Using the chain rule, with $D$ denoting the Jacobian operator,

$$\nabla_W \ell(x, y; f(W)) = (D_W f(W))^\top \nabla_{f(W)} \ell(x, y; f(W)) = (D_W f(W))^\top \nabla_{\bar{W}} \ell(x, y; \bar{W}).$$

We are left with the computation of the Jacobian. We proceed with chain rule

$$f(W) = f_1(f_2(f_3(W))) \cdot W, \quad f_1 : t \in \mathbb{R} \mapsto t^{-1}, f_2 : e \in \mathbb{R}^d \mapsto \|e\|, f_3 : W \in \mathbb{R}^{d \times d} \mapsto W e_x.$$

$$\begin{aligned}
D_W f(W)^\top &= \nabla_W (f_1 \circ f_2 \circ f_3)(W) W^\top + f_1(f_2(f_3(W))) \cdot I \\
&= \frac{-\nabla_W (f_2 \circ f_3)(W)}{f_2(f_3(W))^2} W^\top + \frac{1}{\|W e_x\|} \cdot I = \frac{-f_3(W)(D_W f_3(W))}{\|f_3(W)\| \|W e_x\|^2} W^\top + \frac{1}{\|W e_x\|} \cdot I \\
&= \frac{-W e_x e_x^\top}{\|W e_x\|^3} W^\top + \frac{1}{\|W e_x\|} \cdot I = \frac{1}{\|W e_x\|}(I - \bar{W} e_x e_x^\top \bar{W}^\top).
\end{aligned}$$

This proves the formula written in the main text.

## B EXPERIMENTAL DETAILS

### B.1 MAXIMAL PARAMETERS UPDATES

In order to carefully choose step-sizes that scale well with width $d$ in optimization algorithms, we follow Yang et al. (2021) and consider learning rates consistent with maximal feature learning updates. Here we consider the following initializations:

- $W$ is initialized as a Gaussian random matrix with $\mathcal{N}(0, \frac{1}{d})$ entries.
- Input embeddings $e_x$ and output embeddings $u_y$ are initialized as either random on the unit-sphere in $d$ dimensions, or with Gaussian $\mathcal{N}(0, \frac{1}{d})$ entries. In both cases, every embedding has norm $\approx 1$.

**Updates to $W$.** The updates to the matrix $W$ look as follows:

- SGD with step-size $\eta_W$:

$$W' = W + \eta_W \delta W, \quad \delta W = \sum_j \alpha_j u_{y_j} e_{x_j}^\top,$$

with $\alpha_j = \Theta_d(1)$, and a dimension-independent number of elements in the sum. Choosing $\eta_W = \Theta(1)$ then ensures that for any input embedding $e_x$, we have $\|W'e_x\| = \Theta(1)$ as desired.

- Adam (idealized here as signSGD) with step-size $\eta$:

$$W' = W + \eta_W \operatorname{sign}(\delta W), \quad \operatorname{sign}(\delta W)_{ij} = \frac{\delta W_{ij}}{|\delta W_{ij}|}.$$

The coordinates of $\operatorname{sign}(\delta W)$ are now $\Theta(1)$ instead of $\Theta(1/d)$, thus the step-size needs to be taken as $\eta_W = \Theta(1/d)$ in order to satisfy $\|W'e_x\| = \Theta(1)$ (see (Yang et al., 2021; Yang & Littwin, 2023) for more details)

**Updates to embeddings.** The updates to embeddings look as follows:

- SGD updates:

$$u'_y = u_y + \eta_u \delta u_y, \quad \delta u_y = \sum_j \alpha_j W e_{x_j},$$

$$e'_x = e_x + \eta_e \delta e_x, \quad \delta e_x = \sum_j \alpha'_j W^\top u_{y_j},$$

with $\alpha_j = \Theta(1)$ and a dimension-independent number of $j$s. Since the algorithm ensures $\|We_{x_j}\| = \Theta(1)$ and $\|W^\top u_{y_j}\| = \Theta(1)$ throughout training, choosing $\eta_u, \eta_e = \Theta(1)$ ensures that these conditions continue to hold after each update.

- Adam/signSGD updates:

$$u'_y = u_y + \eta_u \operatorname{sign}(\delta u_y), \quad (\operatorname{sign}(\delta u_y))_i = \frac{(\delta u_y)_i}{|(\delta u_y)_i|},$$

$$e'_x = e_x + \eta_e \operatorname{sign}(\delta e_x), \quad (\operatorname{sign}(\delta e_x))_i = \frac{(\delta e_x)_i}{|(\delta e_x)_i|}.$$

Since the updates have coordinates of order $\Theta(1)$, in order to ensure that embeddings remain of norm $\Theta(1)$ after each update, we thus need $\eta_u, \eta_e = \Theta(1/\sqrt{d})$.

## B.2 ADDITIONAL FIGURES

Our theory predicted optimal scaling laws in $d^{-1+\alpha}$. However, there are some catches behind the proof:

- The lower bound is true when $d \ll N = 100$, otherwise the error can actually reach zero when $d$ becomes larger than a tipping number $d_t$ which compares to $N$. This fact was illustrated on Figure 3. Increasing $N$ augments the tipping point $d_t$, rectifying the learning curve as illustrated on Figure 9.
- This was proven for models where $q(x, y) = q(x)$, and where $q(x)$ is not optimized with respect to $f_*(x)$. As such, it is not clear if those lower bounds hold for optimization-based algorithms, although we argue that we do not expect different mechanisms to take place in the proofs. We illustrate this empirically in the left of Figure 12.

Similarly, the unreasonable effect of learning the embeddings would be highly disappointing if those were hard to optimize in practice. The right of Figure 12 illustrates how with a few steps, one can achieve a zero generalization error when learning the embeddings.

In order to better understand gradient updates, Figure 14 shows the dynamic of the association memory $W$ updated with SGD and a large step size. To validate the approximation (20), Figure 4 plots the generalization error associated with SGD and its theoretical approximation, while Figure 5 illustrates the idealized association scheme $q_\gamma$ associated with a step size $\gamma$, batch size one and a Zipf law on $x \in [N]$.

In order to understand the effect of Adam, we compare it with plain SGD and SGD with rescaled variance on population data. That is, we consider gradient descent with $\nabla_W \mathcal{L}(W)$ (16). The rescale variance SGD, consists in dividing the gradient by the variance of $\nabla_W \ell(X, f_*(X); W)$ (17) when

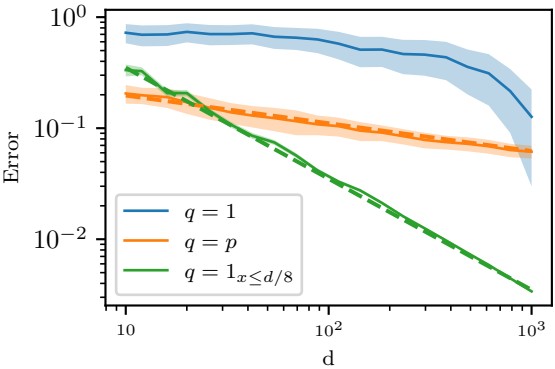

**Figure 9:** Same figure as the right one of Figure 3 yet with a bigger $N$, here $N = 1000$. The dashed curves represent $\mathcal{E} = .35 \cdot d^{-1/4}$ (orange) and $\mathcal{E} = 3.5 \cdot d^{-1}$ (green). They validate the scaling predicted by theory where we used $N = +\infty$ to get tight polynomial scalings of $\mathcal{E}$ (5) with respect to $d$.

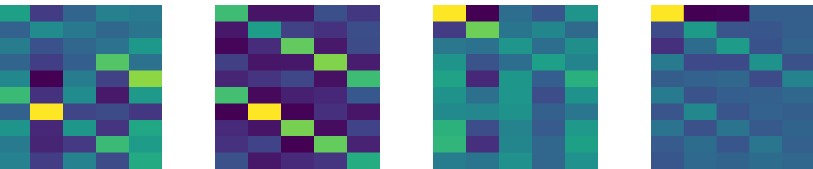

**Figure 10:** Representation of the weight matrix $(u_y^\top W e_x)_{y,x} \in \mathbb{R}^{M \times N}$ for $N = 10$, $M = 5$, $f_*(x) = x \bmod. M$. The data $x$ follows a Zipf-law with $\alpha = 1$ and $T = 10^3$. The matrix $W$ is obtained according to (4) together with the scheme (11). Left: $\rho = 0$ (10), $d = 10$, there is not enough memory capacity, and the model does not succeed to store memories, leading to a large generalization error. Middle left: $\rho = 0$ (10), $d = 50$, there is enough memory capacity, we learn the right association $y = x \bmod. M$. Middle right: $\rho = 1$ (11), $d = 10$, the weighting $q$ allows to store the most important memories beside having a small memory capacity. Right: $\rho = 1$ (11), $d = 50$, the weighting $q$ is too strong which does not allow to store memory associated with rare association (bottom of the matrix).

$X \sim p$ (1). For simplicity, we consider Adam with $\beta_1 = \beta_2 = 0$, in which case, it equates sign SGD, i.e., SGD when considering the sign of each entries of $\nabla_W \mathcal{L}(W)$ in the updates $W_t \to W_{t+1}$. Figures 15 and 16 underpins our intuition that the usefulness of Adam lies in its ability to rescale gradient updates, an effect that could equally be obtained by tuning the learning rate.

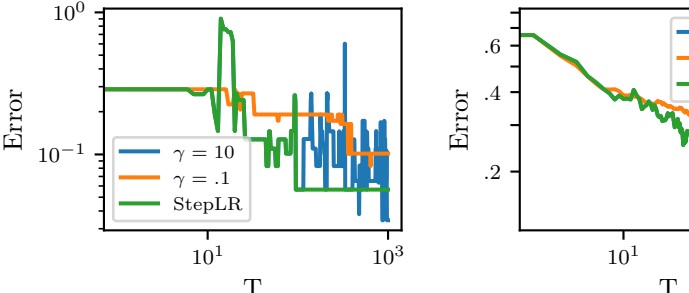

**Figure 11:** Learning curve of the generalization error $\mathcal{E}$ (5) with respect to the number of data processed by stochastic gradient descent in the setting of Figure 6. Left: comparison on a single run. A big step size allows to store more memory at the risk of overwriting past association, which explains the higher variance of the blue curve but its overall better performance. A small step size will avoid loss spikes due to memory overwriting, but will take more time to store rare associations, leading to worse performance. By decreasing the learning rates along training, e.g., with the "StepLR" scheduler (Paszke et al., 2019), one can get the best of both world, i.e., store memories fast at the beginning of training when storage capacity is underused, while being more cautious at the end of training when there is no more "free" memory space. Right: Similar plot with $N = 30$ averaged over one hundred runs.

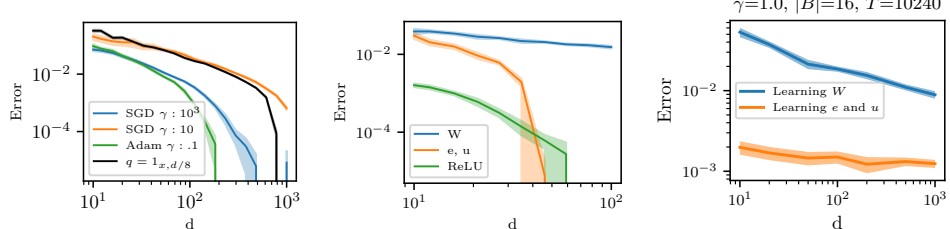

**Figure 12:** Scalings with respect to $d$ for optimization-based algorithms, in the setting of Figure 3. Left: optimization-based algorithms beat the best algorithm designed by hands with $q(x,y) = q(x)$. Note how the curve seems to have the same optimal exponent $\mathcal{E} \asymp d^{-\alpha+1}$ (the left part of the figure show similar slopes for all curves) yet with smaller constant in front, leading to earlier typing point before reaching zero generalization error due to full storage of all the associations. Middle: Comparison of learning the sole matrix $W$ (blue), or learning the embeddings $e$ and $u$ (orange), together with the possibility to use non-linear model $u_y \operatorname{ReLU}(e_x)$ with $e$ and $u$ learned (green). All curves are obtained after $10^3$ updates with batch size $10^3$. Right: Comparison with the same setting as Figure 7. Learning the embeddings or going non-linear allows to impressively optimize memory storage, leading to better exponent with respect to $d$ and earlier tipping point for a fixed number of updates.

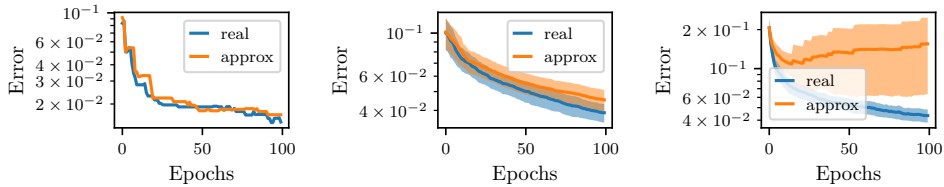

**Figure 13:** Same as Figure 4 yet with batch size equals one thousands $|B| = 10^3$.

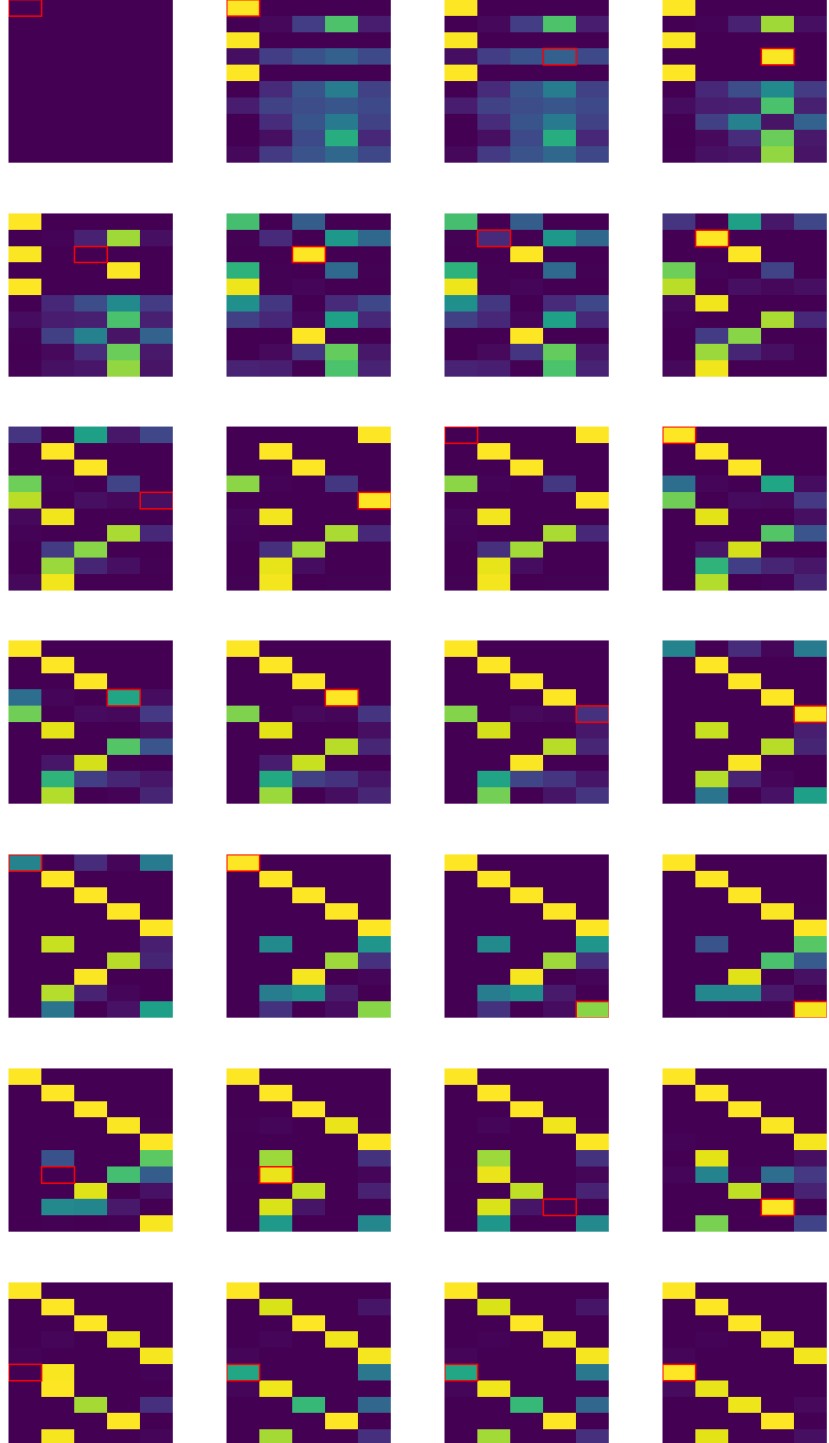

**Figure 14:** Gradient descent dynamics similar to Figure 6 with $d = 10$ and a fixed step size $\gamma = 10$. From time to time, we represent here $t \in \{0, 4, 5, 6, 8, 9, 11, 30, 32, 37, 49, 62, 75, 90\}$, stochastic gradient descent will hit an association that is not properly stored in memory yet (the red boxes). It will consequently update the weight matrix $W_t \to W_{t+1}$ (side by side pairs) to store it. When $d$ is big enough, here $d = 10$, $W$ will end by storing correctly all associations, leading to perfect generalization for future examples.

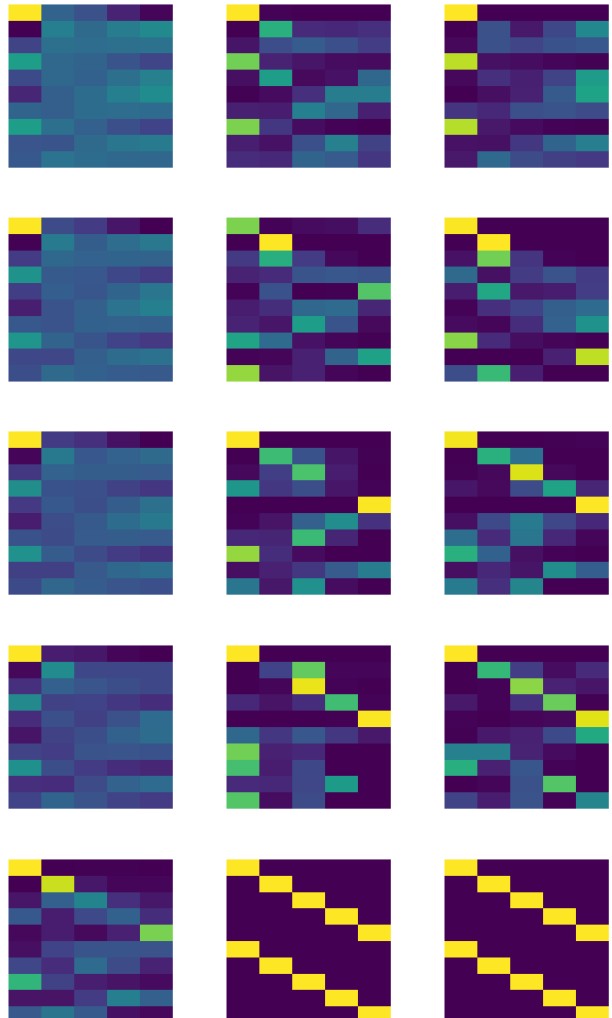

**Figure 15:** Comparison between SGD, signSGD and SGD with normalized variance on population gradient seen from the association matrix $W_t$ at different times in the setting of Figure 14. The different rows correspond to the matrices $W_t$ at time $t \in \{1, 2, 3, 7, 100\}$. Left: Plain SGD. Middle: Adam with $\beta_1 = \beta_2 = 0$, i.e., SignSGD. Right: SGD with normalized variance.

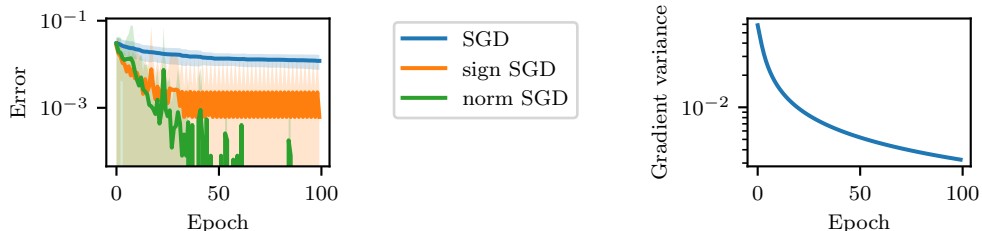

**Figure 16:** Left: Generalization error in the setting of Figure 15. Observe how SGD with rescaled variance (in green), an effect that can be done with SGD after adapting the learning rate, actually performs better than sign SGD (i.e., Adam with $\beta_1 = \beta_2 = 0$). Right: Variance of SGD along the training. As the training goes, SGD is losing momentum due to smaller gradient variances, hence smaller updates.

