# OpenReview forum: "Scaling Laws for Associative Memories"
_ICLR.cc/2024/Conference — ICLR 2024 spotlight_

### Official Review · Reviewer_vzgF · 2023-10-24

**Soundness:** 4 excellent
**Presentation:** 2 fair
**Contribution:** 4 excellent
**Rating:** 8
**Confidence:** 3

**Summary:**

The paper studies scaling laws in a simple, linear associative memory model. The model is aimed at capturing trends in LLMs, which use similar mechanisms. The authors derive scaling laws for the error of the model under varying amounts of data and memory capacity. The results reveal an optimal method of storing information in the model to minimize error. Next, the authors demonstrate that memorizing via gradient updates can be modeled in the framework derived earlier, and show trends of error with respect to learning rate and batch size. The authors finally discuss some additional considerations related to optimization, layer normalization and learned embeddings.

**Strengths:**

**Originality**
The approach taken by this paper is novel, and contrasts with perspectives in prior papers that focus on continuous inputs. What is particularly notable is that the paper considers both errors arising from finite data size and finite capacity and derives relatively explicit expressions for each of these.

**Quality**
Although the proposed model is relatively simple, the theoretical analysis in the paper is extensive as evidenced by Appendix A. The theoretical statements are also backed up by experiments where applicable.

**Clarity**
The paper is mostly adequately well-written. Figures are well illustrated. The notation is well chosen.

**Significance**
The paper appears relatively significant to the field of associative memory. Although it considers a relatively simple memory model, the analysis is quite extensive and could be used in future work. Moreover, connections are drawn to practical LLMs which greatly improves the paper's relevance.

**Weaknesses:**

In my view, the main weaknesses of the paper are related to its clarity. The paper is quite dense with theoretical results, which is good in that the authors provide many contributions. On the other hand, it makes it difficult to interpret and contextualize the results. I would suggest that the authors use more space discussing their results and interpretation, and move some theoretical results to the supplement. One possibility to consider might be adding an extended discussion subsection at the end of each of sections 3 and 4.

Another point of weakness is the description of related work; it would be ideal to significantly expand this section, particularly with respect to the theory on associative memory models. It may be helpful to highlight key results in the associative memory and scaling laws in the related work section (e.g. results on the capacity of other associative memory models, scaling laws for LLMs). This is important to establish the significance of the results in this paper relative to prior work.

One key assumption in the paper is that inputs take discrete values, and that unseen input values lead to errors. It would be helpful to further discuss the realism of the assumption. In particular, when inputs are continuous-valued, we may expect generalization to unseen input values that are similar to previously seen values. When is it (or is not) reasonable to expect this kind of generalization in the discrete setting?

Finally, it would be worth discussing in further detail what the key gaps remain from using the theory developed in this paper to explain scaling in actual, practical LLMs (e.g. what remaining architectural features of LLMs prevent the theory from applying to them).

**Minor Comments**

The placement of figures is sometimes far from where they are referenced in the text

It is unclear what the error margins in figures 3 and 4 represent

The trends in Figure 7 are difficult to interpret due to the variation- it would be ideal to plot an average of many trials

Adding some additional models to Table 1 could be helpful; it might not be worth having a table here if there are only two rows

The log scaling symbol in equation 9 is not formally defined in the main text

**Questions:**

What are the key differences between this work and related work? What are the scaling results for similar memory models that have been previously proposed?

What is the practical significance of having discrete input values? How does this affect how one may consider generalization to unseen inputs?

What key gaps remain between the model considered and practical LLMs?

---

> ### Author Response · Authors · 2023-11-21
> **Rebuttal**
>
> Thank you for your positive review, we appreciate the time you took to help us get the most out of our work.
>
> Your questions relate to the weaknesses you mentioned, and that we will comment now.
> - *"In my view, [...] sections 3 and 4"*:
> Thank you for your suggestions to improve the clarity of our work.
> We will do our best to better contextualize the different results with some extended discussion at the end of Sections 3 and 4.
>
> - *"Another point [...] prior work"*:
> Regarding related work, Hutter derived Proposition 1, and our Theorem 1 is to put in perspective with traditional Hopfield networks that can store $d / \log(d)$ spin patterns, although our setup is slightly different (working with random embeddings).
> Up to our knowledge, no paper has provided a clear understanding of the sample complexity for associative memory, at least not in the realm of LLMs.
>
> - *"One key assumption [...] discrete setting?"*:
> The memorization phenomenon described in the paper is intrinsically of discrete nature.
> It is true that in LLMs, tokens are not completely disjoint unity.
> E.g., if your tokenizer tokenizes "happy" and "happily" as two tokens, their embeddings are probably going to be close, and learning that "happy" is associated with positive sentiments will transfer to "happily".
> This is notably why we were keen to add a paragraph on learning the embeddings (see also Appendix A.8).
> We believe that those phenomena will somehow decrease the "effective number" of tokens, but that the described memorization mechanisms will stay the same (similarly to how the study of sample complexity for linear regression can be extended to non-parametric estimation through the notion of effective dimension in Hilbert spaces).
>
> - *"Finally, [...] applying to them)"*:
> We see our work as a module to be reused to comprehend more finely more complex architecture.
> There are three components that seem important to us to generalize this work.
> 1. Hierarchy: what happens when we put many associative memories one after the other (e.g., the first ones may memorize low-level semantics, while the last ones may perform more abstract pattern matching)?
>
> 2. Superposition: what happens when those modules are put in parallel (e.g., multi-head) and extract patterns that can be reused differently later by subsequent layers?
>
> 3. Entanglement: how to disentangle transformer representation to be able to track pattern matching mechanism / associative memory? Can we easily erase facts from LLMs memory, like someone's birthdate, or the knowledge of Harry Potter, based on manual interventions of the weights?
>
>
> Thank you for your other comments:
> - We will try to remedy the situation regarding the *placement of figures*.
> - In *Figure 3*, the randomness comes from the random embeddings, in *Figure 4*, it also comes from the random data. We will clarify it.
> - The right of *Figure 7* is averaged over several runs in order to showcase the trends.
> - We were hesitant for *Table 1*, some of us believed that it makes sense to keep it really simple as a minimal form of take-home message.
> - Thanks for pointing out the *undefined symbol*. We will define it.
>
> Thank you very much for the great suggestions, and the many comments that will help us improve our work.

---

### Official Review · Reviewer_fK8b · 2023-10-27

**Soundness:** 4 excellent
**Presentation:** 3 good
**Contribution:** 3 good
**Rating:** 8
**Confidence:** 2

**Summary:**

The paper presents an extensive rigorous theoretical analysis of the associative memory capacity of simplified transformer layers. Associative memories are formalized as cross products of input and output tokens, which are assumed to be associated deterministically in the analysis. These associations are combined in the key matrix of an attention layer with hard argmax attention. The analysis is carried out assuming that the input tokens follow a Zipp law, which is commonly observed in naturalistic data.

Finally, the paper contains a numerical analysis of SGD leaning in these layers and it provides some recommendations for Transformers training.

**Strengths:**

I must start by stating that I am not very familiar with the kind of proofs given in the paper and I did not have the time to study them in detail. Therefore, my judgment is conditional on the validity of the statements.

I found the analysis to be useful as it offers a detailed theoretical view on an essential component of modern language models. While the paper makes several simplifications, I do think that the resulting model captures several of the main components of commonly used attention layers.

All in all, I do think that research of this kind is highly needed toi bridge the gap between our understanding of language models and our ability to use them. While this paper is just a small step in this direction, I do think that it is a much needed one. In particular, I highly appreciated the focus on token memorization as the phenomenon seems to be behind most of the capabilities of generative models.

**Weaknesses:**

- Some of the assumptions are rather strong and it is therefore unclear if the insights will generalize to more realistic scenarios. In particular, deterministic associations are rare in real data.

- There is some evidence on the importance of lower weighted components of the attention blocks in the performance of Transformers, which are entirely ignored in the hard argmax model.

- While I do think that the theoretical analysis is insightful, I am not sure that the result of the SGD experiments on the simplified model can cast much insight on actual Transformer training. In fact, the recommendation of small batches and larger step sizes seem to be the opposite of what is known to work in large architectures.

**Questions:**

- Is it possible to extend the analysis to probabilistic associations?
- Is it possible to analyze the softmax model, or is the hard softmax assumption central to the tractability of the model?

---

> ### Author Response · Authors · 2023-11-21
> **Rebuttal**
>
> Thank you for your appreciation of our work, and your great questions to extend our first steps toward a better understanding of modern language models.
>
> Let us first address your comments on the weaknesses.
>
> - *"Some of the assumptions [...] in real data"*"
> We agree that deterministic associations are rare in real data.
> Happily, the proof technique behind our main theorem, Theorem 1, still applies to noisy associations, although the derivations would be more lengthy.
>
> - *"There is some evidence [...] model"*:
> There are definitely second-order effects that we did not succeed to quantify in this paper.
> Typically, lower weighted components could introduce "positive interferences" that improve the performance of the model, but our analysis only considers "worse-cases" when dealing with interferences between memories (see, e.g., paragraph "upper bound" on page 16, or Appendix A.8.1).
>
> - *"While I do think [...] large architectures."*
> We understand your concern about take-away generality.
> We hope that thinking in terms of memorization capacity could help design optimal optimization parameters for more complex models, although we agree that the conclusions are likely to vary from our single-layer model. Nonetheless, note that our conclusions regarding small batches and large learning rate are about SGD, while Adam (the method of choice in practice) does work well with large batches as commonly used in practice. The non-linear nature of the Adam updates makes it more difficult to study precisely, but we hypothesize that its benefits for large batches are due to its pre-conditioning, clipping, and normalization effects, as discussed in Section 4.1.
>
> Regarding your questions:
>
> - *"Is it [...] probabilistic associations?"*
> Yes, it is possible, although it would require lengthier derivations that are beyond the scope of this paper.
>
> - *"Is it [...] the model?"*
> Yes, it is possible to analyze the softmax layer.
> Indeed, in our simple model replacing the hard max by a softmax convexifies the problem, and one can use tools from convex analysis to do so.
> This is a choice we initially considered, although we decided to focus on scaling laws with respect to the 0-1 classification loss, which implies the use of a hard max.

---

### Official Review · Reviewer_fGhC · 2023-10-31

**Soundness:** 3 good
**Presentation:** 2 fair
**Contribution:** 2 fair
**Rating:** 6
**Confidence:** 3

**Summary:**

The authors provide scaling laws for the error of a specific model of associatice memory (it takes inputs x and predicts outputs y, which deterministically depends on x) in terms of the strategy of the construction of this models parameters, the number of input-output pairs seen, and the distribution of the input tokens.
They further experiment how optimized, rather than prescribed weights, relate to the error scaling recovered in the theory. They investigate how several specific architectural and optimization choices affect this error in practice.

---- Update ----
Thanks to the authors clarifications during the rebuttal, my confusion got cleared up. I now understand the paper to be not only an interesting theoretical contribution, but also one that belongs in this context. During our discussion we converged on the points that lead to my misunderstanding, and the authors intend to improve some aspects in the camera-ready version. In the light of this, I improved my score and think that modulo the changes the authors promise, the paper should be accepted.

**Strengths:**

- The derived results look interesting in the context of the chosen model and construction of its parameters.
- Looking at discrete data with a real-word-like distribution is a promising idea.
- I think re-framed in the correct context, the result could add nice insights to scaling laws, even though in their current presentation they are more confusing than insightful.

**Weaknesses:**

- While the introduction and title suggest that the paper considers the memory capacity of associative memories, it seems that in fact it is investigating the error scaling laws of a specific learning problem, where a discrete input x determines an output y. The suspicion that this is learning, is corroborated by the  fact that giving more data for a fixed dimension (e.g. Fig.3 right) improves the error. If the model was truly memorizing, eventually there would be a cut-off and no new data could be taken up by the model for a fixed dimension d (as there is in Hopefield networks, the original 'associative memories'). Under the present title and introduction I would expect scaling laws of the memory capacity in terms of the input parameters, and this is not what the paper is giving. This is the main weakness of the paper; that the motivation, theory and experiments do not form a coherent line of arguments which improve understanding of associative memories and their memorization capacity.
- I find it difficult to comment on the results of the paper in the light of this mismatch, for me, the stated goal to investigate "[...] how different key elements in the training of a transformer influence storage in our memory model." which motivates the experimental section, is not answered at all.
- I want to note that I would be happy to read a rebuttal about why the authors believe their theoretical and empirical analysis is connected to memory capacity as discussed in Figure 2 - it could be that I am missing a piece. Otherwise, I think the results, stated differently, could still be useful to the community, but this would require a complete revision of the paper's motivation and contextualization.

**Questions:**

Abstract
- 'We derive precise scaling laws with respect to sample size and parameter size,' -> it seems there is a subejct missing "We derive precise scalings laws of quantity XY with respect to ...."

Section 1
- It would be nice to give an example of a 'behaviour' of models that can be accessed with scaling laws.
- what is the criterion to qualify a scaling law as 'improved'?
- what is exactly meant by a 'statistical rate' in the present context?
- 'theoretical schemes' -> theoretical predictions?
- 'based on specific' -> 'for specific'?

Section 2
- 'number of data *samples* '
- 'The first/second ones' seems like a wrong english construction of mixing singular and plural.
Section 3
-  as is the case at initialization -> of a neural network/transformer?
Section 4
- m = 5. -> M = 5?

Figures
Fig 5 batch one -? batch size one?
Fig 8 is it SignGD, Adam, or SGD in the plots?

---

> ### Author Response · Authors · 2023-11-21
> **Rebuttal**
>
> Thank you very much for your review.
> We are happy that you find our results insightful and interesting.
> We are sorry to hear that you did not relate with the framing of our results in terms of statistical learning bounds, instead of memory capacity results more common in the neural computation literature. We will do our best to clarify how these viewpoints relate to each other.
> Your comments will help us improve our work, and increase its reach.
>
> Let us address the three weaknesses you are mentioning.
> We hope that our answer will help you better appreciate our work.
>
> -*"I want [...] contextualization"*:
> Figure 2 was actually useful for us to engineer the proof of Theorem 1 (see for example Eq. (30) page 14 in the supplementary materials).
> The mechanism regarding Hopfield networks you are mentioning is exactly the one we are deriving: when $d$ is smaller than the number of associations to memorize, there are interferences between memories, and you can not store all of them.
> Theorem 1 was useful to later predict the empirical observations in Section 4 (as we will detail in point 2).
>
> -*"While the introduction [...] capacity"*:
> It seems that there is some confusion regarding the right of Figure 3.
> The x-axis is the embedding dimension (i.e., the memory capacity), not the number of samples, it shows that as the memory capacity increases, the error decreases.
> The memory cut-off can be seen on the left and middle of Figure 3: as T, the number of data augments, the level lines of the generalization error do not change (the color do not change as we go up) after a certain threshold with respect to $T$ that is linear in $d$.
> This cut off can also be seen on the right of Figure 1.
> Note that in addition to the number of data $T$, the scaling laws are given as a function of $d$ which parameterized the sole parameter of the model, which is the $d$-by-$d$ matrix $W$ Eq. (2) (itself defined through $q$, Eq. (4)).
> Hence one simply has to take $d = \theta^{1/2}$ to cast our scaling into one with respect to the number $\theta\in\N$ of parameters of the model.
> We will make those points clearer in the next revision.
>
> -*"I find it difficult to comment [...], at all"*:
> In order to understand how the key elements in the training of a transformer influence storage in our memory model, we perform some gradient update approximations (which were controlled by experiments on Figure 4), and derive the memorization scheme $q$ that arises from different choice of optimization hyperparameters, plugging this into the theorems of Section 3 allows us to predict the behavior observed on Figure 8: e.g., large learning rates are better, small batch size is better; as well as Figure 7: scheduling is useful when reaching memory overflow.
> Those insights are what we meant by "illustrating how different key elements in the training of a transformer influence storage in our memory model".
>
> Thank you for pointing many typos:
> - The *scaling laws* are the scaling laws of the generalization error (we will write it explicitly).
> - By *"behavior"* we mean the generalization error, and to a certain extent "emerging properties" such as mastering grammar, syntax, translation, reasoning... (lower generalization error in language modeling means better next tokens predictions, which implies the understanding of more phenomenon, and is associated with "emerging properties"). We will make it clearer.
> - *"Improved scaling law"* means that the generalization error is decreasing faster to zero (which can be read in a bigger exponent in the power law of the generalization error with respect to the scaling parameters, i.e. the number of data, or the embedding dimension).
> - *"Statistical rates" means the average (over inherited randomness) rates of convergence of the generalization error towards zero.
> - *Figure 5* represents Adam with $\beta_1 = \beta_2 = 0$, which is equivalent to SignSGD (https://openreview.net/forum?id=S1EwLkW0W).
> - Thank you for spotting all the other typos!
>
> We hope that our answer will help you better appreciate our work, and stay at your disposal to answer further concerns.

---

> > ### Comment · Reviewer_fGhC · 2023-11-22
> > **Answer to rebuttal**
> >
> > Dear authors,
> > I appreciate your answer and clarifications, which helped with my frustrations of grasping the papers underlying structure and reasoning.
> >
> > 1. After reading the other reviews and your comments, I think I am now less confused. I would like to ask, how far you agree with my following understanding of your work, and specifically, which of those aspects of my understanding can still be improved:  Let us assume we have a data distribution of N tokens and a deterministic target function f*. The straightforward thing to do to save this data, is to memorize the data completely (which is exactly what associative memories are in my understanding of its definition). Now since there is the function f*, you do not need to recall every pair individually, but rather you can memorize the f and then apply it to your input to obtain the target y. This should save some space. A classical learning question would be: How many samples do you need to obtain a good generalization error? Now, because transformers have a special outer product weight structure (W), you want to argue about types of learning that set the weights of a transformer matrix to learn f*. A good W would have a low generalization error, while at the same time using only a few parameters d. Instead of looking at the learning dynamics first, you look at several ways of fixing W, and comparing how the generalization error scales in terms of the number of data seen, and the dimension of the model. The way associative memories enter into the discussion is the way in which you fix W, using outer products of random embeddings, which it is akin to the way in which classical associative memories are constructed. With the added component, that since you want to learn a function rather than entries, you use a type of associative memory weight construction that maps an input to an output. From this you derive the scaling laws of the generalization error in terms of d and T. After looking at the scaling laws that govern these specific types of weights set manually, you compare those to weights learned by optimization of losses, and derive some connections on the different properties of the optimization processes, informing practical scenarios.
> > 2. I think the main reason for my previous confusion was the definition of “memory capacity”.  I understand it as the number of words you are able to store in a model (d/log d in the Hopfield). It seems, when you use the word in your work, you often refer to the size of the available space for saving the data (using number of parameters d). This can be justified by the fact that the memory capacity for saving x,y pairs capacity scales as a function of d (even though not directly in d), so you use these terms interchangebly. Do I understand this correctly? I think I was mainly searching for a connection between this number of pairs you can save and the generalization error in the work. Is it then correct to say, that as you increase the number of examples you save, you approximate f* better, but at the same time d limits the number of samples you can save at all. This trade-off is then expressed in the scaling laws you derive.
> > 3. Overall, I agree with reviewer vzgF: the work could benefit from a clearer description and more textual clarifications, to contextualize the theory. Especially the introduction is kept rather short at 1 page and listing a more detailed contribution list with clear connections/ I retract my opinion that the paper would need to be completely rewritten. I think now that I seem to understand the story, it makes a lot of sense and is a very nice contribution. I do not want to take myself as the example of an average reader, but I do think that presented more clearly and with more careful definitions of the words used, I could have gotten to my understanding at least a little bit faster and your work might reach more people that way too.
> > 4. Again, I might be missing something obvious, but I am wondering the following: Is there an intuition on how the complexity of the function f* comes into play?
> >
> > With this, I am very much inclined to change my score. However, I would appreciate it if you were to verify my understanding in point 1, and maybe even give a short description of how my feedback and the feedback of reviewer vzgF would help you to improve the presentation. Given that there is not much time left in the rebuttal period, I don’t expect more than a quick answer. Thank you already!

---

> > > ### Author Response · Authors · 2023-11-22
> > > **Discussion**
> > >
> > > Thank you very much for your quick and helpful answer.
> > >
> > > Your points 1. and 2. are indeed a correct understanding of our work, we are glad this discussion was helpful to identify our misunderstandings, and apologize for the confusion. We will do our best to clarify these points.
> > >
> > > Let us first answer your point 4, which is related to the math behind our work.
> > > - *"Again, [...] Is there an intuition on how the complexity of the function $f_\*$ comes into play?"*:
> > > Because we are focusing on discrete data, we can consider the worse case for $f_*$, i.e. we do not have to restrict the complexity of $f_*$.
> > > The fact that we are considering the worse case for $f_*$ appear in Theorem 2 "there exists a conditional distribution", which is actually the worst possible one.
> > > In the best case $f_*$ is constant, and it can be learned with one sample, which actually appear in the definition of $Q_\infty$ in Eq. (8) or in the $p_*$ of Eq. (9).
> > > However, the difference between an "average" $f_*$ (e.g., $f_*(x) = x mod 5$ as used in our simulations) and the "worst" one would mainly appear as a multiplicative constant in front of the generalization error, but would not change the "scaling law".
> > >
> > > Regarding clarity, we are planning to make the following changes (hopefully we will push a first set of draft changes by the end of the rebuttal period):
> > > 1. Add a diagram that explains the setup (as suggested by reviewer cFvv), so that the skimming-through reader has clear visual clues to understand the setup.
> > > 2. Be more mindful of our wording regarding "scaling law" so that it appears clearly that we are after the generalization error.
> > > We will equally be more thoughtful on how we are referring to the parameter $d$.
> > > We saw $d$ as a "model size/capacity" parameter, which happens to be related to the "token memorization capacity"; we agree that the wording "memory capacity" needs to be better explained to avoid confusion.
> > > 3. We will better relate our work to the literature on associative memories. Although we derived the proofs from scratch, we ended up recovering the $d/log(d)$ of Hopfield memory networks. Yet, as you mentioned, our setting is slightly different since we are memorizing the association $x \to y$ (instead of patterns in $\{0, 1\}^d$).
> > > 4. We will better explain the dichotomy between seeing a transformer as a big memory module, or as the composition of many memory modules (which justified Section 4, since we want to understand how such a module is learned with SGD).
> > >
> > > Thanks again for your highly constructive and positive feedback.
> > > We highly value papers where the reader can easily skim through to get the big picture without having to spend hours figuring out the details.
> > > We are grateful for the time you took to clearly rephrase our paper, which will help us reach a higher level of clarity.

---

> > > > ### Comment · Reviewer_fGhC · 2023-11-23
> > > > **Increased Score**
> > > >
> > > > Thank you a lot for the quick answer, I adapt my original review accordingly!

---

> > > > > ### Author Response · Authors · 2023-11-23
> > > > > **Quick update**
> > > > >
> > > > > We pushed a first set of changes few hours ago.
> > > > > We removed what used to be Figure 7 to gain space, and made the following changes to answer our previous 1-4 points.
> > > > >
> > > > > 1. We added Table 1, which should help get the setup right.
> > > > > 2. We hope that this table will make clear that we are after the scaling law of the generalization error with respect to the parameter d and T.
> > > > > We also changed “memory capacity” for “model capacity” wherever it was confusing.
> > > > > 3. We added a paragraph after Proposition 3 to discuss the relation between our result and Hopfield network. We plan to make further connections with this stream of research in next revisions.
> > > > > 4. We added some sentences at the end of Section 3 to make the dichotomy clearer.
> > > > >
> > > > > Thank you for increasing your score!

---

> > > > > > ### Comment · Reviewer_fGhC · 2023-11-23
> > > > > >
> > > > > > Thank you, I think this already helps a lot!

---

### Official Review · Reviewer_qN1v · 2023-11-01

**Soundness:** 3 good
**Presentation:** 4 excellent
**Contribution:** 3 good
**Rating:** 8
**Confidence:** 4

**Summary:**

The authors explore the behavior of a simple model for associative memory as a weighted sum of outer products of query and key vectors for tokens (matrix). In particular they provide bounds for its generalization error as the number of tokens and the encoding vector size varies and for different choices of the weights in the sum (memory scheme).

More specifically they provide scaling laws in the case memory or data is infinite (respectively for finite data or memory) and memory performance characteristics for weights that are constant or seen-data specific (frequencies). They also study optimization based learning of memorization and how training choices and hyperparameters as in transformers affect its characteristics.

**Strengths:**

- The presentation is excellently organized, the notations, definitions and associated propositions and theorems are carefully stated and accompanied by clean supporting simulation plots, the cases explored make up a comprehensive and complete narrative for this interesting theoretical work.

**Weaknesses:**

- The current setup is synthetic/artificial: it is a drastic simplification of configurations found in practice, e.g. for real transformers. Although there are clear notes in the text for the potential deviations of this simplified model to a real one, it remains to be seen how well analogies hold. To this end, perhaps crisper (albeit riskier) predictions of how some of these results would translate/map to tangible observations in a real transformer would help the reader better appreciate the implications of the theoretical results.

**Questions:**

- For ranges of values for T and d  for data distributions that could map to / feed actual transformers what would be the recommended memory scheme to try in order to minimize generalization error? (This could be a high level and practical direction to the reader who seeks a brief takeaway message).

---

> ### Author Response · Authors · 2023-11-21
> **Rebuttal**
>
> Thank you for your appreciation of our work.
> We are happy to hear that you found our study complete and thorough.
> We appreciate your will to validate our theory with concrete recommendations and predictions of tangible observations.
>
> *"The current setup [...]. To this end, [...] theoretical results."*:
> At the time of writing, we cannot provide you one crisp prediction that is easy to test, but can provide some additional discussion on the matter.
> Here are some practical directions for the reader who wants to take-away our perspective to enlighten their experiments.
>
> 1. Associative memory could be understood with different granularity: e.g., a transformer is seen abstractly as a big associative memory machine, and tokens represent abstract "pieces of knowledge"; but layers can also be seen as associative memories working at finer levels, e.g., remembering attributes of people, such as birth dates.
>
> 2. Knowledge can be entangled.
> For example, to recognize the language of a document, it is probably enough to average all tokens of this document and learn a simple classification rule between this average (the input) and language (the class/output).
> However, some of our experiments suggest that some "interpretable behaviors" (e.g., "people birth dates") can be entangled/distributed across heads or layers.
> Being able to train while enforcing "layer disentanglement" would solve many issues in LLMs.
>
> 3. Once you have a layer working for a sub-task you do not need much to learn to solve new instances of this sub-task. E.g., if you have learned to recognize languages (the sub-task of recognizing languages is useful for next token prediction) through tokens averages (an attention head does the average, and the MLP layer do the association "tokens average"->language); it will be easy learn to recognize new languages with a few gradient update on the MLP to add the new association new average->new language.
>
> Let us now turn those guidelines into a concrete prediction on the minimal curated data and the (curriculum) learning process needed to see emergent reasoning behavior.
> To ground the discussion, let us consider a "piece of knowledge" that corresponds to the ability to prove statements by induction.
> Roughly speaking we can assume that a type of reasoning should have been seen about 30 times for it to be learnable.
> This particularly true when the learner has already been tuned/wired to be able to do variables binding and recognize reasoning patterns, i.e., there is an (entangled) inner transformer layer that has been learned and acts as an associative memory of reasoning patterns (similarly to the language recognition layer described above).
> Hence, we could assume that with proper curriculum learning, 30 proofs by inductions (that are various enough to abstract the induction mechanism, and avoid prediction based on confounding factors) should be enough to learn to prove things by induction (this is one concrete prediction).
>
> *"For ranges of [...] takeaway message)"*:
> To answer your specific question, in general the best memory scheme is the thresholding one where one remembers all the "pieces of knowledge" $x$ that have been seen frequently enough (say more than 30 times to average out noise - in our noiseless setting 1 time is enough).
> When in the presence of limited memory, best is to memorize the most frequent ones.
> Concretely, when having $T$ data, we want to remember all $x$ such that the number of time $x$ has been seen is greater than 30, i.e., $T p(x) > 30$, which can be done by considering a memory capacity $d$ (which, in our paper, is the embedding dimension - but in practice could be defined as some effective dimension) such that $30 = T p(d) \simeq T d^{-1/\alpha}$ (under the Zipf law hypothesis).
> This retrieves the optimal scaling $d \simeq T^{1/\alpha}$ found by equating the two terms in Eq. (15).
> Although in this example, $T$, $d$ and $x$ corresponds to abstraction to be determined in a large corpus of texts processed by a transformer.
>
> We appreciate your will to get the best out of our work, notably through the search of its practical implications, and hope that our response brings more answers than questions, and convince you of the valuable perspectives brought by associative memories to comprehend LLMs.

---

### Official Review · Reviewer_cFvv · 2023-11-02

**Soundness:** 4 excellent
**Presentation:** 3 good
**Contribution:** 3 good
**Rating:** 8
**Confidence:** 2

**Summary:**

This paper performs a study of the scaling laws from the perspective of associative memory, studying the phenomena by formalizing a highly controlled experimental setting. They test this phenomena across the amount of training data and the embedding (memory) dimension of that data and observe scaling law trends that resemble those of LLMs, indicating that the conclusions drawn in this paper will likely extrapolate beyond the scope of small associative memories.

**Strengths:**

## Admirably formalizes the scaling laws in Transformers as a memorization/memory retrieval task in Associative Memories

- (+ +) The paper clearly and thoroughly defines a "sandbox" problem setting where we can study scaling laws (of discrete data domains, like the vocabulary tokens in NLP) using principles of Associative Memory
- (+) The paper includes experiments using the Associative Memory sandbox to draw conclusions about good optimizers, learning rates, and batch sizes in larger models.
- (+) Supplementary includes complete and well-organized code for all experiments in the paper.

**Originality**: I am not aware of existing works studying the scaling laws from the perspective of Associative Memory.

**Quality**: The paper is of high quality, though I did not read the Appendix.

**Clarity**: The paper is very clear, though accessibility to the average reader could be improved.

**Significance**: Medium-High -- this is another work drawing formal connections between foundation models and associative memory, providing theoretical structure to a field designed primarily by empirical results.

**Weaknesses:**

## Experiments not able to scale to large models

1. (-) It took several readings to understand the experimental setup. The clarity of the paper would be improved with a small architectural diagram describing the setting.
2. (-) To my understanding, the proposed method can only study Transformer blocks individually, not the entire Transformer as a whole (This is my understanding of Sec 4 paragraph 1: "our model is a proxy for the inner layers of a transformer")
3. (-) Like 2., the proposed method does not allow words in an input sequence the ability to talk to each other, which is how the attention mechanism in Transformers actually works (see Question 1). Thus, the sandbox is a very limited tool to study larger language models.

**Questions:**

1. Sec 2 par 1:

> "For example, $N$ could be the number of potential sequences of fixed word length in the English language, while $M$ would be all the potential words to complete the sequence"

Unfortunately, there is no modern model that actually treats all possible sequences of a fixed word length as a single token. But a recently proposed method derives the Transformer as an Associative Memory (see [Energy Transformer](https://arxiv.org/abs/2302.07253)). Could you explain how the experimental setup could be adapted to more advanced associative memory structures that contain multiple weight matrices and allow token-token interaction?

---

> ### Author Response · Authors · 2023-11-21
> **Rebuttal**
>
> Thank you for your appreciation of our work.
> We appreciate your curiosity towards the next steps to be taken to perform a more comprehensive study.
>
> - *"It took several [...] setting"*:
> Thank you for your feedback, we will add a diagram to ease readability.
>
> - *"To my understanding, [...] transformer)"* There are two ways to use our result.
> Either one sees a transformer as a big associative memory machine and derives LLMs scaling laws approximation from ours (this is what Hutter, 2021 or Maloney et al., 2022 do, as well as the energy transformer you point out).
> Either one sees our result as describing the behavior of some blocks (which is the future use we argue for).
> We will make this dichotomy clearer.
>
> - *"Like 2., [...] models."*
> To see the relevance of our model for studying interactions between input tokens (as opposed to input-output associations), we refer to [Bietti et al. 2023](https://arxiv.org/abs/2306.00802), where it is shown that gradient dynamics on key-query matrices in a transformer lead to outer-product associative memories of a similar form to the ones we study. In that work, the weighting of different outer products elements also seems related to token frequencies (see e.g., Figure 7 in their paper, which shows how different tokens are more or less prominent in key-query matrices). Studying precise scaling laws for such multi-layer transformers is beyond the scope of the present work, but we expect our work to be generalizable to that setting.
>
> Thank you for the time you devoted to review and help us improve our work.

---

### Author Response · Authors · 2023-11-21
**General answer**

We would like to thank all the reviewers for the time they took to review our work.
We are happy that this work was received with enthusiasm.
Most of you shared our perspective that the statistical understanding of transformers is an important problem, that associative memories might provide useful clues, and that this work is a good first step in this direction.
We are equally happy for the many great suggestions to improve our work, and to study further question, including the impact of noise, of higher-order (lower weights) mechanisms at play, and the description of more intricate models.
Your encouragements give strong motivations to continue this stream of research.

---

### Meta-Review · Area_Chair_aNgz · 2023-12-05

**Metareview:**

The paper presents novel theoretical results on scaling laws in associative memories. The reviewers agreed they are of interest to the community. The reviews also presented a number of comments that should allow the authors to clarify the presentations of their results.

**Justification For Why Not Higher Score:**

I do not have any justification here, more lack of comparison due to limited statistics.

**Justification For Why Not Lower Score:**

The results seem of interest for the wider community.

---

### Decision · Program_Chairs · 2024-01-16

Accept (spotlight)